# Crystal structure and substrate-induced activation of ADAMTS13

Anastasis Petri[1,4], Hyo Jung Kim [2,4], Yaoxian Xu[1,4], Rens de Groot [1], Chan Li[2], Aline Vandenbulcke[3], Karen Vanhoorelbeke[3], Jonas Emsley[2,5] & James T.B. Crawley [1,5]

Platelet recruitment to sites of blood vessel damage is highly dependent upon von Willebrand factor (VWF). VWF platelet-tethering function is proteolytically regulated by the metalloprotease ADAMTS13. Proteolysis depends upon shear-induced conformational changes in VWF that reveal the A2 domain cleavage site. Multiple ADAMTS13 exosite interactions are involved in recognition of the unfolded A2 domain. Here we report through kinetic analyses that, in binding VWF, the ADAMTS13 cysteine-rich and spacer domain exosites bring enzyme and substrate into proximity. Thereafter, binding of the ADAMTS13 disintegrin-like domain exosite to VWF allosterically activates the adjacent metalloprotease domain to facilitate proteolysis. The crystal structure of the ADAMTS13 metalloprotease to spacer domains reveals that the metalloprotease domain exhibits a latent conformation in which the active-site cleft is occluded supporting the requirement for an allosteric change to enable accommodation of the substrate. Our data demonstrate that VWF functions as both the activating cofactor and substrate for ADAMTS13.

[1] Centre for Haematology, Imperial College London, London, UK. [2] Centre for Biomolecular Sciences, School of Pharmacy, University of Nottingham, Nottingham, UK. [3] Laboratory for Thrombosis Research, KU Leuven, Kortrijk, Belgium. [4] These authors contributed equally: Anastasis Petri, Hyo Jung Kim, Yaoxian Xu. [5] These authors jointly supervised this work: Jonas Emsley, James T.B. Crawley. Correspondence and requests for materials should be addressed to J.E. (email: jonas.emsley@nottingham.ac.uk) or to J.T.B.C. (email: j.crawley@imperial.ac.uk)

Von Willebrand factor (VWF) (Fig. 1a) is critical for platelet recruitment at sites of blood vessel damage[1]. VWF exists as disulfide-linked multimers ranging from dimers to ~100-mers that are released by endothelial cells and platelets into the plasma[2]. In circulation, plasma VWF adopts a conformation in which the binding sites for platelet GPIbα in the VWF A1 domain are concealed, allowing VWF to circulate without unwanted platelet binding[3]. Following vessel damage, however, subendothelial collagen is exposed to which globular VWF binds via its A3 domain. Tethered VWF then undergoes a structural transition in response to the shear forces exerted by the flowing blood that exposes the previously hidden GPIbα-binding sites in its A1 domains[3]. This facilitates capture of platelets from blood under high shear leading to formation of the platelet plug. Larger VWF multimers are more hemostatically competent as

they contain more collagen and platelet-binding sites, and also more readily unravel under elevated shear forces. High VWF plasma concentration and large VWF multimer size are risk factors for thrombosis[4–6]. Consequently, VWF multimeric size/platelet-tethering function is proteolytically regulated by the metalloprotease (MP) ADAMTS13[5]. ADAMTS13 is a ~190 kDa multidomain plasma glycoprotein. Low ADAMTS13 is associated with increased risk of myocardial infarction and stroke[4]. Severe ADAMTS13 deficiency is a hallmark of thrombotic thrombocytopenic purpura[4]. Nascent ADAMTS13 has a short propeptide that is proteolytically removed prior to secretion[5,7]. The N-terminal domains of secreted ADAMTS13 comprise the MP, disintegrin-like (Dis), a thrombospondin type 1 (TSP1) repeat, cysteine-rich (Cys-rich), and Spacer domains[5]. The crystal structure of the ADAMTS13 DTCS domains has been resolved

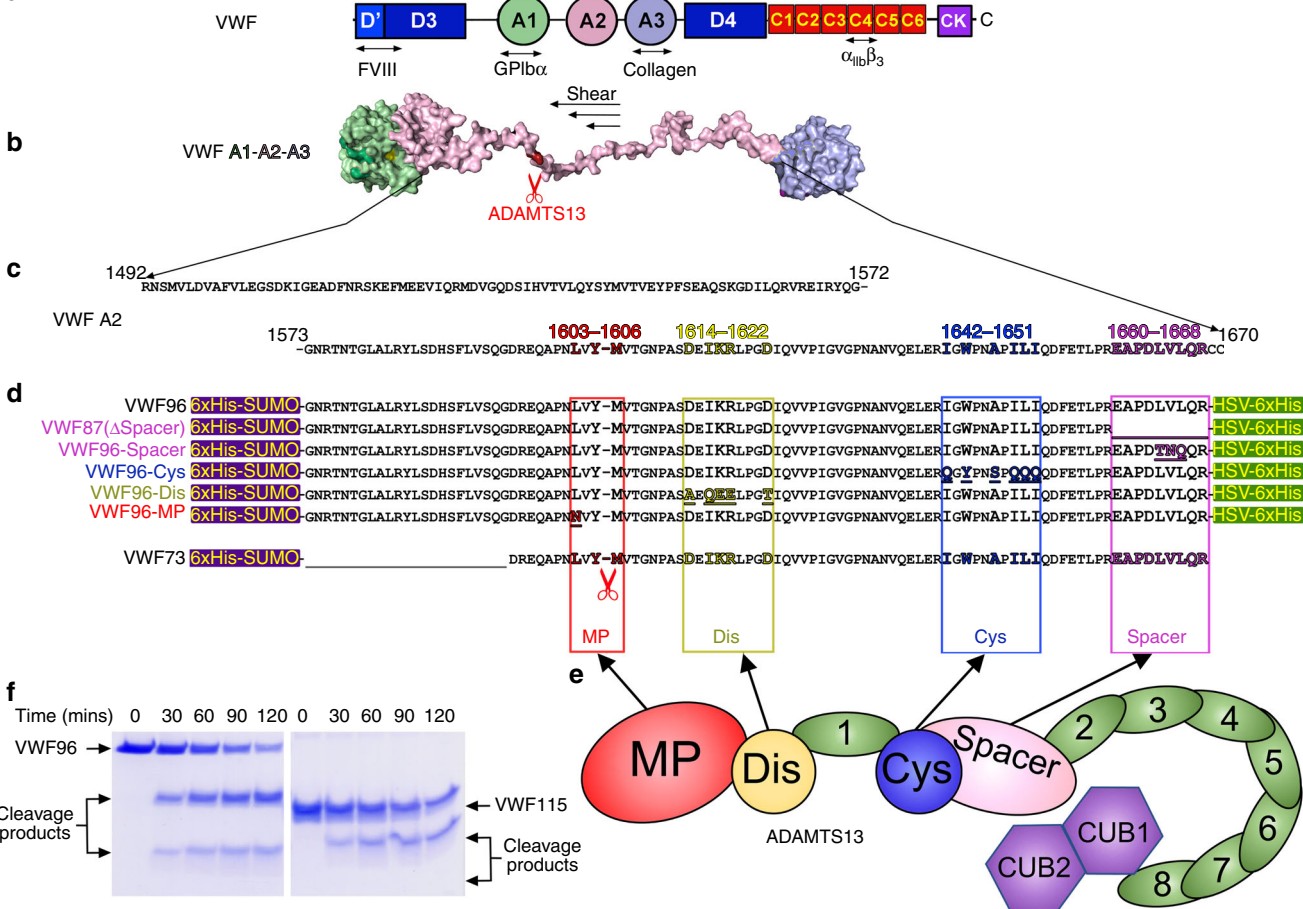

**Fig. 1** Von Willebrand factor (VWF) and its exosite interactions with ADAMTS13. **a** Schematic representation of the domain organization of a VWF monomer. Locations of disulfide bonds involved in VWF multimerization are marked (S–S). Locations of the major ligand-binding sites are shown. **b** Model of the central VWF A domains (A1–A2–A3) is shown following shear-induced unfolding. The Tyr[1605]-Met[1606] scissile bond in the VWF A2 domain that is revealed and cleaved by ADAMTS13 is shown in red. **c** Amino acid sequence of the entire A2 domain (residues Arg[1492]-Cys[1670]). Amino acids previously suggested to be involved in recognition by different ADAMTS13 domains are highlighted (metalloprotease (MP) domain—red, disintegrin-like (Dis) domain—yellow, cysteine-rich domain (Cys)—blue, Spacer domain—pink). **d** VWF A2 domain fragments that were used in this study. VWF96 is a 96 amino acid A2 domain fragment (Gly[1573]-Arg[1668]) with a 13 kDa N-terminal SUMO tag and a short C-terminal HSV tag and that spans the ADAMTS13 cleavage site and each of the Dis, Cys-rich, and Spacer domain exosite-binding regions. With the tags, VWF96 has a MW of ~32 kDa. The regions deleted or mutated in each of the VWF fragments are highlighted. **e** Schematic representation of the domain organization of ADAMTS13. Domains are labelled. The C-terminal tail consisting of the thrombospondin (TSP1) repeats 2–8 and CUB domains is depicted folding back to interact with the central Spacer domain. **f** The proteolysis of VWF96 (32 kDa) by ADAMTS13 is compared to a previous VWF A2 domain fragment (Glu[1554]-Arg[1668]), VWF115 (18 kDa) that lacks SUMO and HSV tags, but contains a short N-terminal Xpress epitope and 6xHis tag. For this 0.5 nM ADAMTS13 was incubated with 3 μM VWF substrate at 37 °C. Sub-samples were taken at specified times and analyzed by sodium dodecyl sulfate-polyacrylamide gel electrophoresis (SDS-PAGE) to visualize proteolysis—note the smallest VWF115 cleavage product is not stained due to its small size. Original gel images are provided in the Source Data file. These data reveal that the identity of the tags does not influence proteolysis

(i.e., without the catalytic MP domain)[8]. The C-terminal domains consist of seven further TSP1 repeats and two CUB domains that fold back and interact with the central Spacer domain[9–12].

There are multiple conformation-dependent exosite interactions between ADAMTS13 and VWF that determine the specificity and timing of proteolysis[13]. VWF circulates in a globular conformation and its central A2 domain is folded such that the cleavage site is buried, making it resistant to ADAMTS13 proteolysis[14]. Only when elevated rheological shear forces induce VWF unravelling (either (i) at sites of vessel damage, (ii) during endothelial secretion, or (iii) during passage through high shear environments) are cryptic exosite-binding sites as well as the $Tyr^{1605}$-$Met^{1606}$ cleavage site revealed in the VWF A2 domain (Fig. 1b, c)[13]. Specific recognition of the unfolded A2 domain is mediated by exosites within the ADAMTS13 Spacer, Cys-rich and Dis domains that are predicted to sequentially interact with amino acids $Glu^{1660}$-$Arg^{1668}$, $Ile^{1642}$-$Ile^{1651}$, and $Asp^{1614}$-$Asp^{1622}$ in VWF, respectively (Fig. 1d, e)[15–24]. These three exosite interactions culminate in the presentation of the scissile bond to the MP domain active-site[13]. The multiple exosite interactions have given rise to the so-called "molecular zipper" model of ADAMTS13 function[13]. The ADAMTS13 MP domain contains a highly conserved 3xHis $Zn^{2+}$-binding motif and a catalytic $Glu^{225}$ residue that form the active-site. Predicted subsites (S1 and S1′ pockets) on either side of the active-site specifically accommodate the VWF P1 ($Tyr^{1605}$) and P1′ ($Met^{1606}$) residues[21]. The ADAMTS13 MP domain also binds the highly important P3 residue ($Leu^{1603}$) in VWF[25]. The MP domain is predicted to contain a functionally important $Ca^{2+}$-binding site adjacent to the active-site, and also an additional double $Ca^{2+}$-binding site[26].

The mode of action of ADAMTS13 is unusual inasmuch as proteolysis is dictated by shear-dependent conformational changes in the substrate (VWF), as opposed to on-demand activation of the enzyme[13]. Classically, plasma protease zymogens (e.g., complement, hemostatic, fibrinolytic proteases) are proteolytically activated upon demand. These enzymes frequently rely upon cofactors to localize or enhance their proteolytic function. Moreover, inhibition of the proteases within each of these enzyme systems, through the actions of the large number of plasma inhibitors, temporally and spatially regulates their function. At face value, ADAMTS13 does not adhere to this mode of action as it: (i) is secreted in a seemingly constitutively active form[7], (ii) lacks an "on-demand" activation step, and (iii) has no cofactor that regulates its function. Despite its apparent constitutive activity, ADAMTS13 exhibits proteolytic specificity for a single site of only one physiological substrate (VWF $Tyr^{1605}$-$Met^{1606}$)[5]. ADAMTS13 is seemingly resistant to the effects of all plasma inhibitors and, consequently, it has a very long active plasma half-life (3–7 days) for a protease, which is, therefore, dictated by clearance (rather than inhibition)[27]. To rationalize how ADAMTS13 exhibits such high substrate specificity despite circulating as an apparently active enzyme, we hypothesized that ADAMTS13 circulates in a latent form that requires allosteric activation mediated by one or more of its exosite interactions with VWF. Using kinetic analysis of substrate proteolysis, we demonstrate that, in binding VWF, the ADAMTS13 Cys-rich and Spacer domain exosites bring the enzyme and the unfolded VWF A2 domain into proximity. Thereafter, engagement of the ADAMTS13 Dis exosite with VWF allosterically activates the MP domain to facilitate proteolysis. Resolution of the crystal structure of the ADAMTS13 MP to Spacer domains, the first such structure for any ADAMTS family member, reveals that the active-site cleft in the MP domain is occluded, suggesting a latent conformation that requires an allosteric change to enable accommodation of the substrate. Our data thus demonstrate that

VWF functions as an activating cofactor for ADAMTS13 that precedes its proteolysis.

## Results

**ADAMTS13 exosites in VWF proteolysis**. Physiologically, ADAMTS13 only proteolyzes the VWF A2 domain once it is unravelled by elevated shear forces. Recognition of the unfolded A2 domain occurs through the ADAMTS13 N-terminal domains (MP through to Spacer—termed MDTCS) (Fig. 1e). Although the VWF amino acids involved in ADAMTS13 recognition and proteolysis have been mapped[18–20,24,25,28], the biochemical contribution of each exosite interaction to proteolysis has not been characterized. To do this, we generated a 96 amino acid VWF A2 domain fragment (residues $Gly^{1573}$-$Arg^{1668}$, termed VWF96) that spans the cleavage site and contains all of the MDTCS exosite-binding regions fused to an N-terminal SUMO tag and a C-terminal HSV tag. Proteolysis of VWF96 by ADAMTS13 occurs in the absence of shear-induced unfolding, like previously characterized ADAMTS13 substrates (Fig. 1f)[29]. We also created VWF96 variants containing mutations or deletions that specifically disrupt the Spacer, Cys-rich, or Dis domain exosite interaction sites (Fig. 1d). Although not strictly an exosite interaction, we also substituted the P3 residue (L1603N) in VWF96, which is important for recognition by the MP domain S3 subsite, to evaluate the contribution of MP domain recognition beyond the cleavage site residues. VWF96 fragments were purified prior to qualitative analysis of their proteolysis by ADAMTS13 (Supplementary Fig 1). As expected, all VWF96 variants (VWF87 (ΔSpacer), VWF96-Spacer, VWF96-Cys, VWF96-Dis, and VWF96-MP) were proteolyzed slower than the wild-type VWF96.

VWF96 proteolysis was monitored kinetically by enzyme-linked immunosorbent assay (ELISA) to quantify the deficiency associated with disruption of each exosite interaction. From time-course reactions, we derived the catalytic efficiency ($k_{cat}/K_m$) of proteolysis: $110.7 \times 10^4\,M^{-1}\,s^{-1}$ (Fig. 2a and Table 1), which is very close to previous estimates for other similar VWF A2 domain fragments[20,24,25]. Reactions containing the VWF96 variants (using higher ADAMTS13 concentrations to accommodate their reduced rates of proteolysis) were analyzed in the same way. Deletion of the residues associated with the Spacer domain exosite interaction ($Glu^{1660}$-$Arg^{1668}$) in VWF87(ΔSpacer) resulted in a 14.7-fold reduction in proteolysis ($k_{cat}/K_m$; $7.51 \times 10^4\,M^{-1}\,s^{-1}$) (Fig. 2b and Table 1). Similarly, substitution of three hydrophobic residues within this same region (L1664T/V1665N/L1666Q) using VWF96-Spacer caused an 8.4-fold reduction in proteolysis, demonstrating the contribution of these hydrophobic amino acids to the Spacer domain exosite interaction (Fig. 2c and Table 1). Disruption of the Cys-rich domain exosite-binding region in VWF96-Cys (I1642Q/W1644Y/A1647S/I1649Q/L1650Q/I1651Q) caused a 15.6-fold reduction in proteolysis, similar to a previous report (Fig. 2d and Table 1)[24]. Mutation of the P3 amino acid (L1603N) in VWF96-MP known to be important for MP domain recognition resulted in 138-fold reduced proteolysis (Fig. 2f and Table 1)[25]. The most profound reduction in catalytic efficiency was observed when the Dis domain exosite-binding region (D1614A/I1616Q/K1617E/R1618E/D1622T) was mutated ($k_{cat}/K_m$; $0.21 \times 10^4\,M^{-1}\,s^{-1}$), which caused a 527-fold decrease in proteolysis (Fig. 2e and Table 1).

**Kinetic analysis of ADAMTS13 exosite functions**. Reductions in catalytic efficiency ($k_{cat}/K_m$) can be caused by increases in $K_m$ (i.e., reduced substrate binding), decreases in $k_{cat}$, reflecting reduced substrate turnover (indicative of reduced functionality of

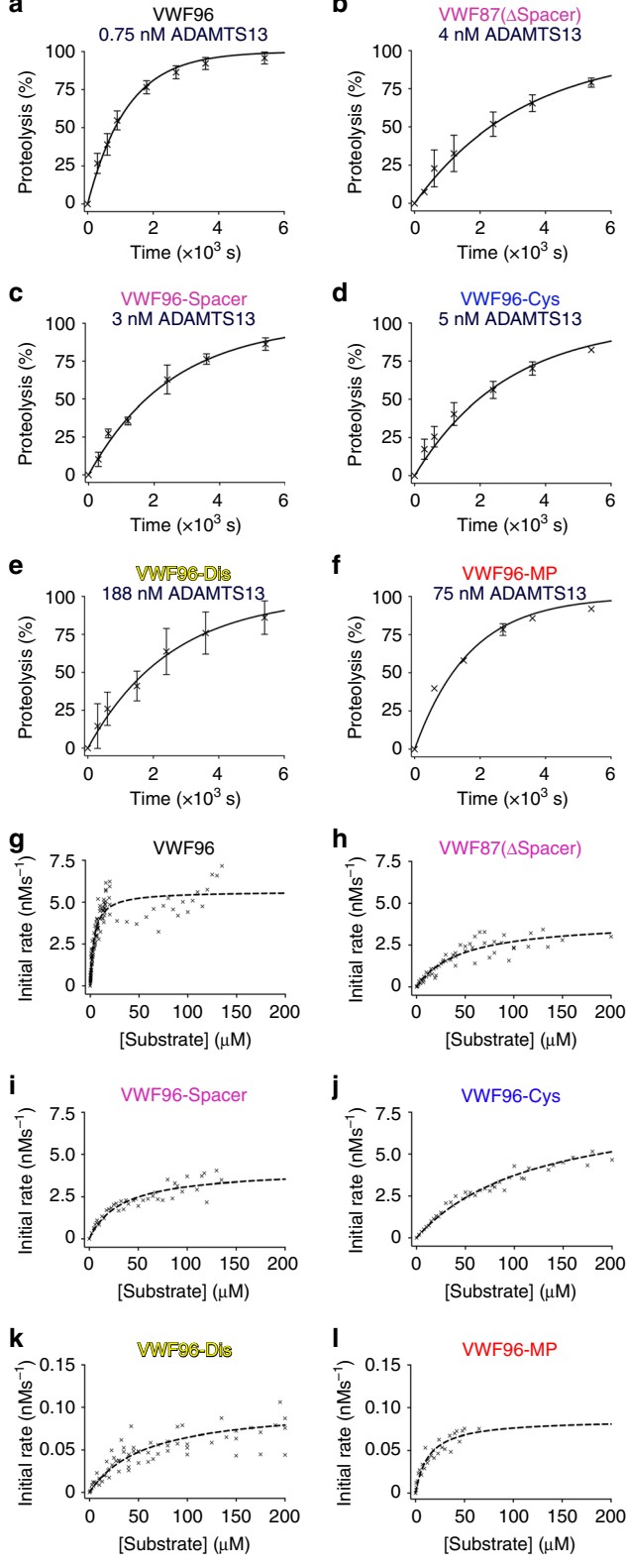

**Fig. 2** Kinetics of Von Willebrand factor 96 (VWF96) variant proteolysis by ADAMTS13. **a-f** VWF96 variants (as labelled) were incubated with ADAMTS13 (concentration is given for each substrate). Sub-samples at different time-points (0–2 h) were stopped with EDTA and proteolysis quantified by enzyme-linked immunosorbent assay (ELISA). **a** For VWF96, 0.75 nM ADAMTS13 was used and 750 nM VWF96. Data plotted are mean ± SD ($n = 16$). **b-f** ADAMTS13 concentrations were increased as noted to accommodate the reduced rate of proteolysis of each VWF96 variant to enable reactions to proceed to completion within 2 h. For all reactions, the substrate concentration used was $<K_m$ for proteolysis. Data presented are mean ± SD, **a** $n = 16$, **b** $n = 4$, **c** $n = 3$, **d** $n = 7$, **e** $n = 8$, and **f** $n = 4$. From these time-course reactions, the catalytic efficiency ($k_{cat}/K_m$) for each substrate was derived—see Table 1. **g-l** ADAMTS13 and VWF96 or VWF96 variants (as labelled) were incubated at 37 °C and proteolysis monitored over time by ELISA. From each progress curve, the initial rate of substrate proteolysis ADAMTS13 nM$^{-1}$ was determined and plotted as a function of substrate concentration. Data were fitted to derive the $V_{max}$ from which the $k_{cat}$ was derived, and also the $K_m$ (concentration of substrate at $V_{max}/2$). From these data, an independent determination of the catalytic efficiency ($k_{cat}/K_m$) for each substrate was derived—see Table 1. Note the different axis scales used for VWF96(Dis) and VWF96(MP) due to the markedly reduced $k_{cat}$ for proteolysis of these two substrates. Number of different substrate concentrations analyzed were, **g** $n = 113$, **h** $n = 58$, **i** $n = 41$, **j** $n = 41$, **k** $n = 84$, and **l** $n = 28$. Raw data underlying all reported averages are provided in the Source Data file

derived $k_{cat}/K_m$ ($141.6 \times 10^4\,\mathrm{M^{-1}\,s^{-1}}$) from these data is in agreement with that from the time-course kinetics ($110.7 \times 10^4\,\mathrm{M^{-1}\,s^{-1}}$).

The reduced proteolysis of VWF87(ΔSpacer) and VWF96-Spacer by ADAMTS13 was associated with increases in $K_m$ (46.1 μM and 32.7 μM, respectively), with little effect upon the $k_{cat}$, indicating that the Spacer domain exosite is involved primarily in substrate binding (Fig. 2h, i and Table 1). Similarly, the reduction in proteolysis of VWF96-Cys was also due to reduced substrate recognition (i.e., 30-fold increased $K_m$, 123 μM) (Fig. 2j and Table 1). Consistent with the proximity of the P3 L1603N mutation to the Tyr$^{1605}$-Met$^{1606}$ cleavage site, the 218-fold reduced proteolysis of VWF96-MP was primarily mediated by a 66-fold reduction in $k_{cat}$ (0.086 s$^{-1}$) (Fig. 2l and Table 1), suggesting that this substitution influences the accommodation of the adjacent cleavage site by the MP domain. Disruption of the Dis domain exosite interaction resulted in a 787-fold reduction in catalytic efficiency ($k_{cat}/K_m$; $0.18 \times 10^4\,\mathrm{M^{-1}\,s^{-1}}$), similar to that obtained from the time-course kinetics ($0.21 \times 10^4\,\mathrm{M^{-1}\,s^{-1}}$). This reduction was due to both a 14-fold increase in $K_m$ (56.4 μM), reflecting a role in substrate binding, and a 56-fold decrease in $k_{cat}$ (0.1 s$^{-1}$) (Fig. 2k and Table 1). The relative effects of each exosite interaction are best appreciated when all data are plotted on the same axes (Supplementary Fig. 2).

The changes in both $k_{cat}$ and $K_m$ associated with disruption of the Dis exosite interaction are indicative of an interaction site that alters the function of the active-site. This may reflect a mechanism in which the engagement of the ADAMTS13 Dis domain exosite with VWF allosterically modulates the functionality of the active-site in the adjacent MP domain.

**ADAMTS13 exosites in VWF A2 domain binding.** To corroborate the decreases in substrate binding reflected by the increases in $K_m$ for VWF96 variant proteolysis, we performed binding assays using ADAMTS13 (Fig. 3a). As previously reported, the $K_{D(App)}$ derived for ADAMTS13 binding to an immobilized A2 domain fragment from these binding assays

the active-site), or a combination of both. To determine how each exosite interaction influences VWF proteolysis, we derived the individual constants, $k_{cat}$ and $K_m$, through analysis of the initial rate of proteolysis measured as a function of substrate concentration (Fig. 2g–l)[20,21,29]. For VWF96, we derived a $k_{cat}$ of 5.65 s$^{-1}$ and a $K_m$ of 3.99 μM, which are in close agreement with previous estimates (Fig. 2g and Table 1)[20,21]. The independently

**Table 1 Kinetic constants for proteolysis of VWF96 and VWF96 variants by ADAMTS13**

| Substrate | $k_{cat}/K_m \pm$ SEM[a] ($\times 10^4$ M$^{-1}$ s$^{-1}$) | Fold decrease[a] ($\downarrow$) | $k_{cat} \pm$ SEM[b] (s$^{-1}$) | Fold difference[b] ($\uparrow/\downarrow$) | $K_m \pm$ SEM[b] ($\mu$M) | Fold increase[b] ($\uparrow$) | $k_{cat}/K_m \pm$ SEM[b] ($\times 10^4$ M$^{-1}$ s$^{-1}$) | Fold decrease[b] ($\downarrow$) |
|---|---|---|---|---|---|---|---|---|
| VWF96 | 110.7 ± 2.28 | – | 5.65 ± 0.16 | – | 3.99 ± 0.41 | – | 141.6 ± 15.22 | – |
| VWF87(ΔSpacer) | 7.51 ± 0.39 | 14.7↓ | 3.97 ± 0.25 | 1.42↓ | 46.1 ± 7.45 | 11.6↑ | 8.60 ± 1.49 | 16.5↓ |
| VWF96-Spacer | 13.1 ± 0.56 | 8.45↓ | 4.11 ± 0.28 | 1.37↓ | 32.7 ± 6.23 | 8.20↑ | 12.6 ± 2.55 | 11.2↓ |
| VWF96-Cys | 7.11 ± 0.26 | 15.6↓ | 8.31 ± 0.53 | 1.47↑ | 123.4 ± 14.0 | 30.9↑ | 6.73 ± 0.92 | 21.0↓ |
| VWF96-Dis | 0.21 ± 0.01 | 527↓ | 0.101 ± 0.006 | 55.9↓ | 56.4 ± 10.2 | 14.1↑ | 0.18 ± 0.03 | 787↓ |
| VWF96-MP | 0.80 ± 0.03 | 138↓ | 0.086 ± 0.006 | 66.0↓ | 13.3 ± 2.79 | 3.33↑ | 0.65 ± 0.14 | 218↓ |

The catalytic efficiency ($k_{cat}/K_m$) of proteolysis for each VWF96 variant was derived from time-course assays (a) that monitored proteolysis of each variant over time (Fig. 2a–f). The fold decrease in proteolysis for each VWF96 variant is shown relative to VWF96—downward arrow denotes decreases; upward arrow denotes increases. From the Michaelis–Menten kinetics (b) monitoring the initial rate of proteolysis as a function of substrate concentration (Fig. 2g–l), the individual constants, $k_{cat}$ and $K_m$, were derived and the independent derivation of the $k_{cat}/K_m$. Fold changes for each constant relative to VWF96 are shown. Raw data underlying all reported averages are provided in the Source Data file corresponding to the data from Fig. 2a–l
VWF Von Willebrand factor, MP metalloprotease, Dis disintegrin-like, Cys cysteine-rich
[a]Data derived from time-course assays
[b]Data derived from Michaelis–Menten kinetics

($40 \pm 1.3$ nM) is approximately two orders of magnitude lower than the $K_m$ for proteolysis ($4 \mu$M)[20,21,29]. This likely reflects changes in the conformation of VWF96 when it is adsorbed onto the plate surface, making it more uniformly permissive to binding ADAMTS13. In solution, the VWF96 fragment may be in equilibrium between open and closed states, with only a small proportion (perhaps 1–10%) of VWF96 at any one time being permissive for higher affinity binding, resulting in the discrepancy between the $K_m$ and $K_{D(App)}$ values. Despite these differences, the dependency upon the exosites in the N-terminal domains of ADAMTS13 for both substrate binding and proteolysis is very similar. Deletion of the Spacer domain exosite-binding region caused a 12-fold increase in $K_{D(App)}$ ($475 \pm 61$ nM) (Fig. 3a). For VWF96-Cys there was a 16-fold increase in $K_{D(App)}$ ($632 \pm 41$ nM), similar to the 15-fold increases in $K_{D(App)}$ for VWF96-Dis binding ($615 \pm 52$ nM). Mutation of L1603 in VWF96-MP resulted in limited effects upon $K_{D(App)}$.

To analyze solution-phase binding between ADAMTS13 and the VWF A2 domain fragment, we employed isothermal titration calorimetry using inactive ADAMTS13 MDTCS(E225Q) and tag-free VWF73 (Fig. 1). From the isotherms, we derived a solution-phase $K_D$ of 450 nM, which is closer to the $K_m$ for proteolysis than the $K_{D(App)}$ derived from the plate-binding assays (Fig. 3b). The isothermal profile for the binding of VWF73 to MDTCS (E225Q) revealed a favorable enthalpic component, but unfavorable entropic contribution (Fig. 3c).

**Crystal structure of 3H9 Fab-MDTCS complex**. Recombinant-inactive MDTCS was expressed in insect (S2) cells; however, crystallization with this sample was not successful. As the crystal structure of the ADAMTS13 DTCS fragment lacking the MP domain had previously been resolved[8], we hypothesized that MDTCS did not crystallize due to flexibility of the MP domain. We therefore utilized the Fab fragment of an inhibitory anti-ADAMTS13 MP domain monoclonal antibody (3H9) to restrict MP domain flexibility[30]. We isolated the 110 kDa 3H9 Fab-MDTCS complex and crystallized this with the structure determined to a resolution of 2.8 Å (Supplementary Table 1). The overall structure of 3H9 Fab-MDTCS complex viewed from the VWF-binding face is illustrated in Fig. 4a. The 3H9 Fab interacts exclusively with the MP domain—predominantly via interactions of the Fab heavy-chain CDR1 (residues 24–33), CDR2 (residues 52–57), and CDR3 (residues 99–118) loops engaging the MP domain at sites that likely overlap with the S1 and S3 selectivity pockets (Supplementary Fig. 3).

The overall structure of the ADAMTS13 MP domain is stabilized by multiple disulfide bonds and bound $Zn^{2+}$ and $Ca^{2+}$ ions, making it unlikely that the inhibitory Fab binding alters the local main chain conformation significantly. Moreover, the conformation of the ADAMTS13 MP when compared to other ADAMTS family MP domains (discussed below) reveals that there are no major differences in the overall structure; however, it is possible that Fab binding induces subtle changes in the positioning of the flexible loops or local changes in the side chain rotamers.

**Structure of ADAMTS13 MDTCS**. The "front view" of MDTCS (without the 3H9 Fab) is presented with the active-site and the functional exosites highlighted (Fig. 4b). In this orientation, the active-site and each of the Dis, Cys-rich, and Spacer domain exosites align on the same face of the molecule. Observed from the "top" (Fig. 4c), MDTCS is "butterfly shaped" with the MP and Spacer domains forming two "outer wings" separated by ~40 Å. The Dis and Cys-rich domains form the two "inner wings", which are separated by the first TSP1 repeat. We detected N-linked

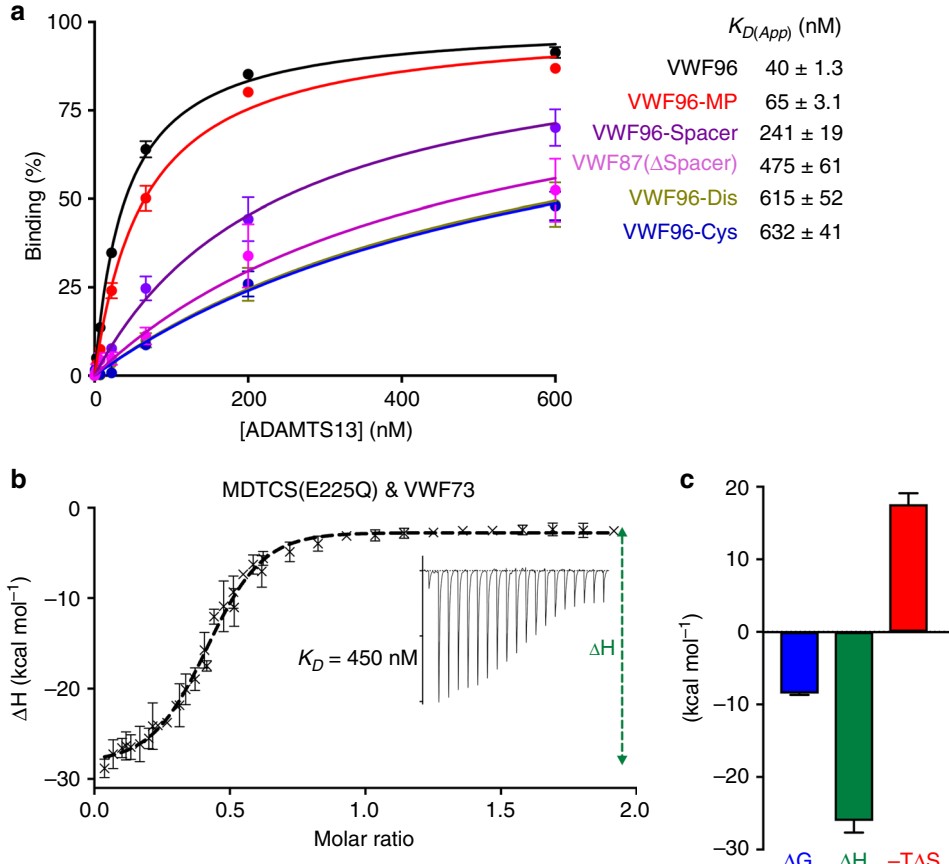

**Fig. 3** Binding of ADAMTS13 to Von Willebrand factor (VWF) A2 domain fragments. **a** Plate binding of ADAMTS13 to VWF96 variants. VWF96, and variants thereof, were adsorbed onto 96-well microtiter plates and incubated with increasing concentrations of ADAMTS13 (0–600 nM) for 1 h. Wells were washed and ADAMTS13 binding detected using an anti-ADAMTS13 TSP2–4 polyclonal antibody. Each concentration was assayed in duplicate on each plate and the data presented are mean ± SEM of three separate experiments. Data were fitted and the $K_{D(App)}$ derived for ADAMTS13 binding to each VWF96 variant, which are given. **b** Solution-phase binding between ADAMTS13 MDTCS(E225Q) and VWF73 was measured by isothermal titration calorimetry (ITC). Twenty-seven micromolar MDTCS(E225Q) active-site mutant in 20 mM HEPES (pH 7.5), 150 mM NaCl, and 5 mM $CaCl_2$ was placed into both sample and control cells. Repeated 0.4–2 µl injections of either tag-free VWF73 (279 µM) in 20 mM HEPES (pH 7.5), 150 mM NaCl, containing 5 mM $CaCl_2$ (into the sample cell) or 20 mM HEPES (pH 7.5), 150 mM NaCl, and 5 mM $CaCl_2$ alone (control cell) were performed and heat changes monitored from which changes in Gibbs free energy ($\Delta G$), enthalpy ($\Delta H$), and entropy ($\Delta S$) were derived as a function of VWF73 concentration. The data presented are the mean ± SEM of four separate experiments. From the plotted data, the solution-phase binding affinity between MDTCS(E225Q) and VWF73 was derived ($K_D$; 450 nM). An example of the raw data is depicted in the inset. **c** MDTCS(E225Q) binding to VWF73 is associated with an unfavorable entropy change. The data presented are the mean ± SEM of four separate experiments. Raw data underlying all reported averages are provided in the Source Data file

glycans on all previously identified sites within MDTCS (Asn[142] and Asn[146] in the MP domain, Asn[552] in the Cys-rich domain, and Asn[579], Asn[614], and Asn[667] in the Spacer domain)[31]. In addition, we also resolved an O-linked glycan on Ser[399], but not on any of the other previously reported sites[31]. When compared to the structure of the ADAMTS13 DTCS domains (PDB ID 3GHM [https://doi.org/10.2210/pdb3GHM/pdb])[8], the Dis domain is "twisted" by ~30° and "declined" by ~7 Å relative to the Cys-rich domain (Fig. 4d and Supplementary Movie 1). This is due to differences in the conformation of the central TSP1 domain, which reduces the distance between Dis and Cys-rich domain exosites by ~6 Å when compared to the DTCS structure.

As previously observed[8], the Dis and Cys-rich domains are topologically related involving an α-helix, 3–4 irregular loops, and a pair of antiparallel β-strands (Fig. 5). The Dis domain is shorter (47 amino acids) than the Cys-rich domain (67 amino acids), which also contains a $3_{10}$ α-helix between $L2_C$ and $L3_C$ loops. Loops are denoted as L1-3$_D$ and L1-4$_C$ according to

the morphology. Although the overall topology of the Dis and Cys-rich domains is alike, loop arrangements and sizes vary and they do not superpose well (root-mean-square deviation (RMSD) = 2.54 Å, 52 aligned residues). The previous DTCS structure revealed an RMSD value of 1.82 Å (54 aligned residues) when the Dis and Cys-rich domains were superposed[8]. In common with the previous DTCS structure, loop L1$_D$ (also termed the V-loop[8]), which likely forms part of the Dis domain exosite, and three loops in the Cys-rich domain are poorly defined in the electron density indicative of a high degree of flexibility in these domains (Fig. 5).

The electrostatic potential at the molecular surface of MDTCS reveals that the substrate-binding face of the MP-Dis involves a ring of positively charged residues Arg[326], His[328], Arg[349], Arg[370], Arg[372] from the Dis and Arg[257] from MP with a depression at the center formed by hydrophobic residues Leu[351] and Pro[353] (Fig. 6). The distance between the active-site $Zn^{2+}$ ion and Arg[349] in the Dis domain exosite is 26 Å. This is in good agreement with the 26 Å distance between the VWF Tyr[1605]-Met[1606] scissile bond (that must lie across the active-site) and Asp[1614] (that interacts

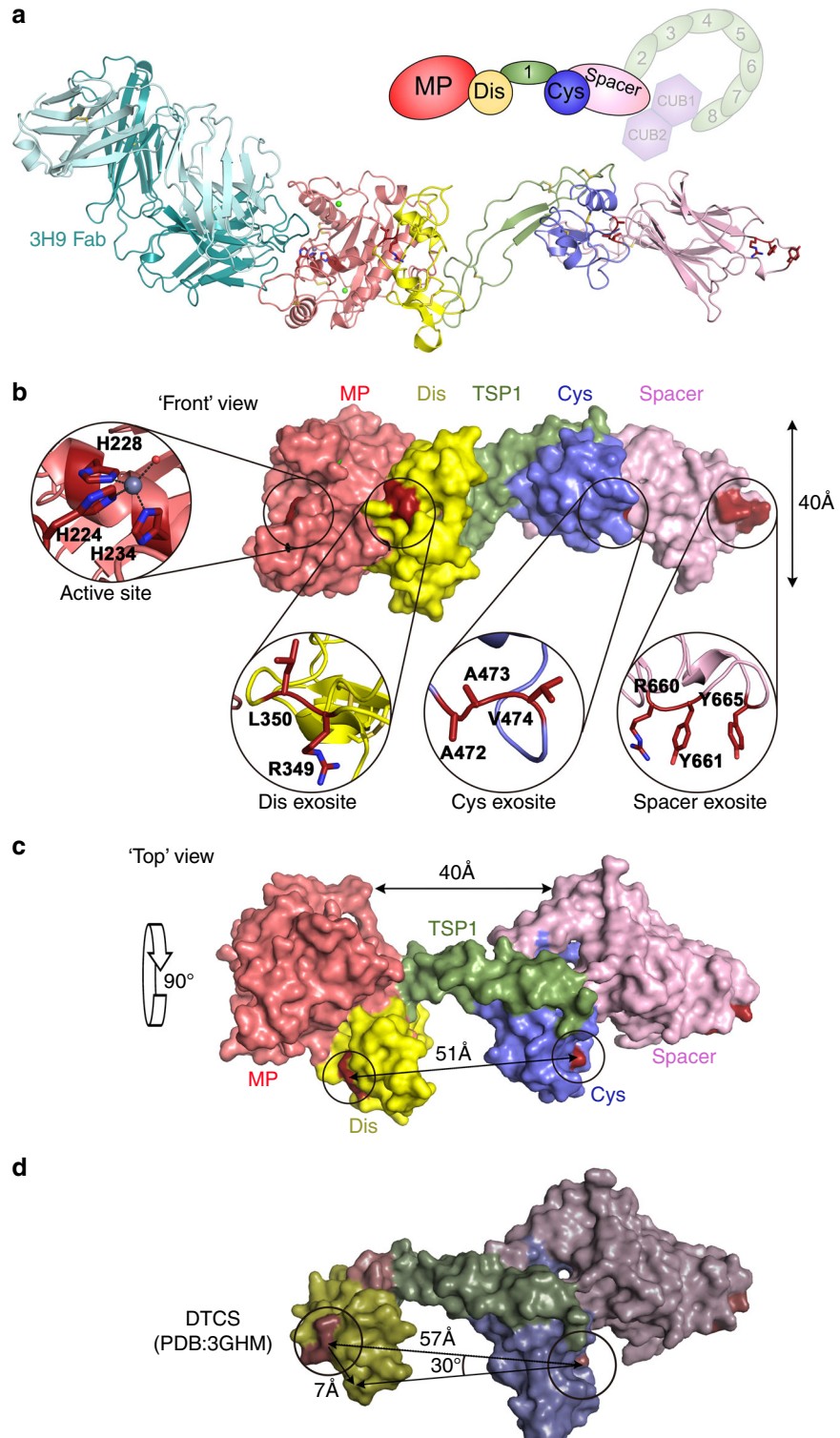

**Fig. 4** Crystal structure of 3H9 Fab-MDTCS (MP through to Spacer) complex. **a** Cartoon representation of ADAMTS13 MDTCS domains bound to the inhibitory 3H9 Fab fragment (teal) (PDB ID: 6QIG). ADAMTS13 domains are labelled and color-coded according to the schematic representation above. **b** Surface structure of ADAMTS13 MDTCS domains (with the Fab removed) viewed from the "front"—substrate-binding face. The ADAMTS13 active-site and exosites are highlighted with residues implicated in the function of each exosite labelled in the insets. Note the linear alignment of the metalloprotease (MP) domain active-site and disintegrin-like (Dis), cysteine (Cys)-rich, and Spacer domain exosites. **c** Surface structure of ADAMTS13 MDTCS domains (with the Fab removed) viewed from the "top"—rotated forward by 90° from **b**. MDTCS exhibits a butterfly conformation with the MP and Spacer domains forming "wings" separated by 40 Å. The Dis and Cys-rich domain exosites are separated by a distance of 51 Å. **d** Surface structure of the previously determined DTCS domains (3GHM [https://doi.org/10.2210/pdb3GHM/pdb]). The notable difference between the two structures (apart from the presence/absence of the MP domain) is the location of the Dis and Cys-rich domain exosites, which in our structure is twisted by ~30° and declined by ~7 Å, which reduces the distance between Dis and Cys-rich domain exosites by 6 Å—see Supplementary Movie 1

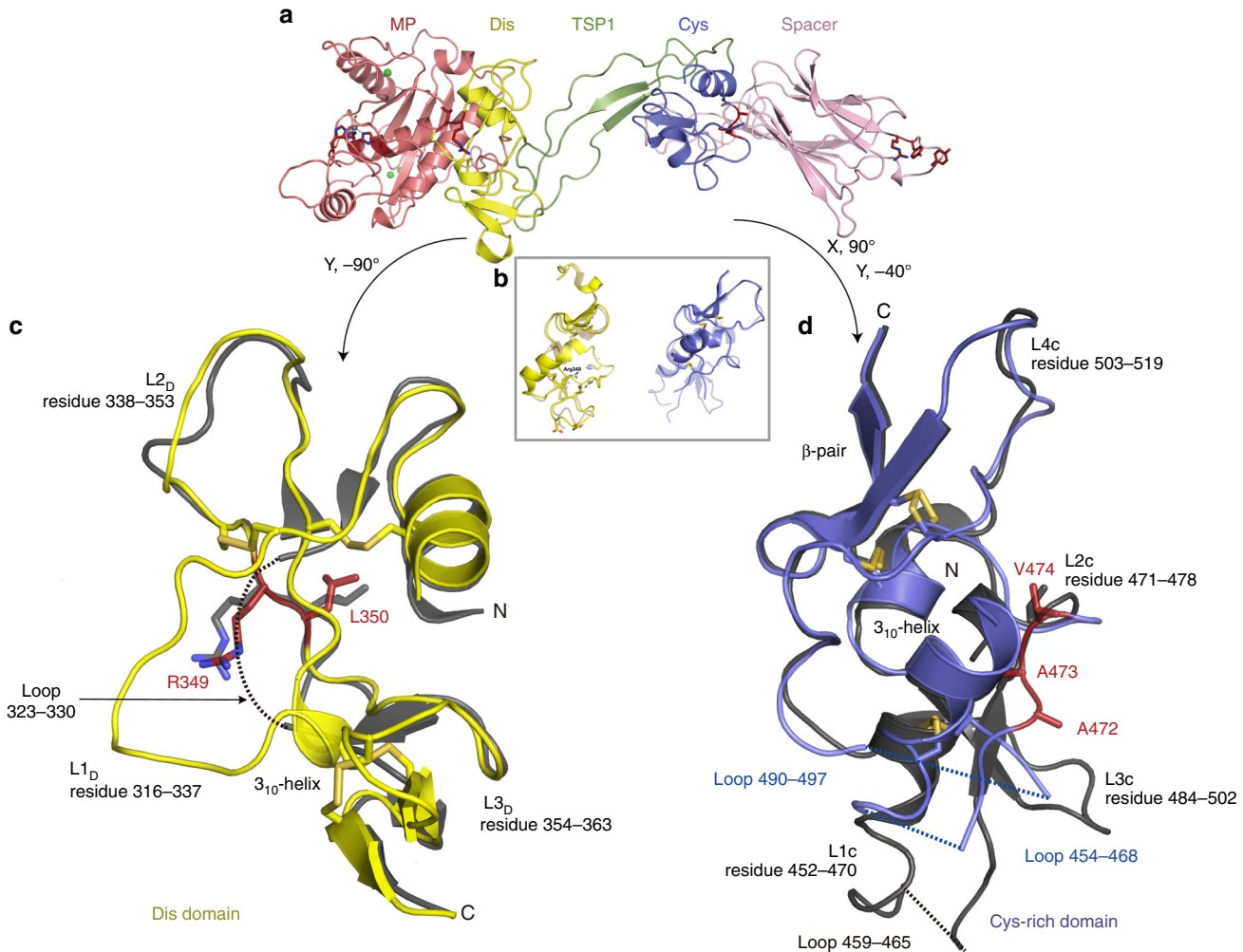

**Fig. 5** Comparison of the disintegrin-like (Dis) and cysteine (Cys)-rich domain structures. **a** Ribbon representation of "front" view of MDTCS (MP through to Spacer) with domains labelled/colored. **b** Dis (yellow) and Cys-rich (blue) domains are shown in the same orientation to reveal their structural homology overlaid on the previous ADAMTS13 DTCS domain structure of these domains. **c** Dis domain (yellow) overlaid on the previous structure of this domain (gray) with the loops L1-3$_D$ labelled. Dis domain exosite residues Arg[349] and Leu[350] are shown as sticks and highlighted in red. L1$_D$ residues 323–330 were missing in the previous DTCS structure of this domain—dotted line. L1$_D$ is connected to L2$_D$ (Cys[322]-Cys[347]) and a β-strand (Cys[332]-Cys[366]), L2$_D$ has additional connection with the α-helix (Cys[311]-Cys[337]), and L3$_D$ is linked with the other β-strand by Cys[360]-Cys[371]. **d** Ribbon representation of ADAMTS13 Cys-rich domain (blue) overlaid on previously determined DTCS domain structure (gray). Loops L1-4$_C$ are labelled. Cys-rich domain exosite residues Ala[472]-Val[474] are shown as sticks and highlighted in red. Unlike L1$_D$, L1$_C$ and L2$_C$ are free of disulfide bonds and there is a small α-helix at the end of L2$_C$. This small 3$_{10}$ helix is linked with the β-strand by a disulfide bond (Cys[483]-Cys[522]). L3$_C$, connected with the main α-helix by Cys[450]-Cys[487], starts from the end of small α-helix and extended to L4$_C$. L4$_C$ has a link with the β-strand by Cys[508]-Cys[527]. Unresolved residues from the DTCS structure (3GHM [https://doi.org/10.2210/pdb3GHM/pdb]) are denoted by dotted lines

with Arg[349] in the Dis domain[20]) when this peptide is arranged in an extended conformation (Fig. 6). Arg[326] and His[328] contribute to the overall positive charge of L1$_D$, whereas Asp[340] and Asp[343] are responsible for a negatively charged patch on the surface of L2$_D$ (that spans the previously termed hypervariable region[8]). The VWF A2 sequence that harbors the Dis domain exosite-binding region, involving Asp[1614], Glu[1615], Lys[1617], Arg[1618], and Asp[1622] shows negative and positive residues that may interact with the complementary charges in L1$_D$ and L2$_D$ (Fig. 6).

The main interaction between Cys-rich domain and VWF A2 is hydrophobic in nature involving Gly[471]-Val[474] (Cys-rich) and Ile[1642]-Ile[1651] (VWF)[24]. In ADAMTS13, hydrophobic residues cluster on the surface involving Leu[443], Met[446], Ala[472], Ala[473], Val[474], Leu[482], Met[486], Met[509], Leu[620], and Leu[621] at the interface between the Cys-rich and Spacer domains. The Spacer domain consists of nine antiparallel β-strands and is very similar to the

previous structure of this domain[8]. Two β-sheets are linearly aligned, composed of 5 and 4 β-strands, respectively. Hydrophobic residues point towards the cleft between the two sheets, and all charged residues (except Glu[641]) are directed towards the solvent interface. A long loop (Arg[660]-Asp[672]) (also containing two very short helices) harbors several residues of functional significance. Arg[659], Arg[660], Tyr[661], and Tyr[665] within this loop were originally suggested to be important for the interaction with VWF Glu[1660]-Arg[1668][22,23]. However, more recently, these residues have been implicated in the interaction of the Spacer domain with the C-terminal CUB domains that fold back to form a "closed" conformation of the full-length molecule (Fig. 1e)[9,10]. It thus seems likely that the Spacer domain residues that interact with VWF do not overlap with this region. Furthermore, given our data revealing the functional importance of the hydrophobic VWF residues Leu[1664], Val[1665], and Leu[1666] in binding the Spacer domain exosite,

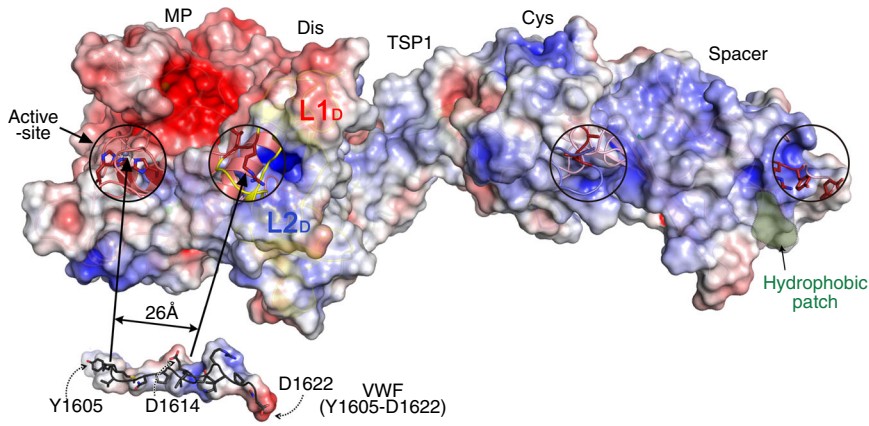

**Fig. 6** Electrostatic potential at the molecular surface of ADAMTS13 MDTCS (MP through to Spacer). The charged surface representation of MDTCS is presented in the "front" view as in Fig. 4 b (negative charges—red; positive charges—blue). Domains are labelled above. The active-site and the disintegrin-like (Dis), cysteine (Cys)-rich and Spacer domain exosites are circled. The L1$_D$ loop in the Dis domain that was not fully resolved in the DTCS structure is highlighted, and also the L2$_D$ loop. Together, these loops provide the interaction site for reciprocally charged Von Willebrand factor (VWF) residues 1615–1622. The peptide Tyr$^{1605}$-Asp$^{1622}$ is shown. The distance from the metalloprotease (MP) domain active-site to Arg$^{349}$ in the Dis domain is 26 Å, which matches the distance from the Tyr$^{1605}$-Met$^{1606}$ scissile bond to Asp$^{1614}$ that interacts with Arg$^{349}$. The Spacer domain exosite involving Arg$^{660}$, Tyr$^{661}$, and Tyr$^{665}$ has been reported to interact with the C-terminal CUB domains of ADAMTS13. We propose that the Spacer domain makes a hydrophobic interaction with Leu$^{1664}$, Val$^{1665}$, and Leu$^{1666}$ with an adjacent surface hydrophobic patch (green) centered around Ile$^{611}$

we propose that these residues may form a hydrophobic interaction with a hydrophobic pocket centered around Ile$^{611}$ on the substrate-binding face of the Spacer domain (Fig. 6).

**Structure of the ADAMTS13 MP domain.** The ADAMTS13 MP domain contains the HEXXHXXGXXH MP motif and a conserved Met-turn (Met$^{249}$), which are hallmarks of this family of proteases[32]. The Zn$^{2+}$ ion is coordinated directly by the side chains of His$^{224}$, His$^{228}$, and His$^{234}$, adjacent to the active-site Glu$^{225}$, which is substituted to Gln in our inactive variant (Fig. 7a, b). Three bound Ca$^{2+}$ ions are resolved in the MP domain. The first Ca$^{2+}$ ion (Ca1) is coordinated by the negatively charged side chains of Asp$^{182}$ and Glu$^{212}$ and the carbonyl backbone of Leu$^{183}$, Arg$^{190}$, and Val$^{192}$. This functionally important Ca$^{2+}$-binding site is located adjacent to the active-site cleft and plays a structural role in maintaining the shape of the active center (Fig. 7b)[26]. Previously, Asp$^{187}$ had been implicated in the coordination of this Ca$^{2+}$ ion, but our structure reveals that this amino acid protrudes at the outer edge of the Ca$^{2+}$-binding loop and likely fulfills a different functional role in proteolysis[26]. Two additional Ca$^{2+}$ ions form a double Ca$^{2+}$-binding site on the rear side of the MP domain relative to the active-site. Asp$^{166}$, Asp$^{173}$, Asp$^{284}$, and Glu$^{83}$ coordinate these two Ca$^{2+}$ ions (Ca2 and Ca3) (Fig. 7b). MP domain residues 276–305 form a 30 amino acid irregular loop that connects the Dis domain and is stabilized by connection with the β5- and β3-strands through the side chain of residue Asp$^{284}$ coordinating the Ca$^{2+}$-binding site (Ca3).

The sequence identity between the ADAMTS13 MP domain and these other ADAMTS family members is 29% (ADAMTS1), 33% (ADAMTS4), and 29% (ADAMTS5) and superimposition of the ADAMTS13 MP domain structure with those of ADAMTS1 (PDB ID: 3Q2G [https://doi.org/10.2210/pdb3Q2G/pdb]), ADAMTS4 (PDB ID: 4WK7 [https://doi.org/10.2210/pdb4WK7/pdb]), and ADAMTS5 (PDB ID: 3LJT [https://doi.org/10.2210/pdb3LJT/pdb]) structures reveals RMSD values of 1.19Å$^2$, 1.10Å$^2$, and 1.10Å$^2$, respectively (Fig. 7c)[33–35]. In this comparison, ADAMTS13 is structurally distinct to ADAMTS1, 4, and 5 in a number of areas; (i) to the right of the Zn$^{2+}$ ion, three

charged residues Arg$^{193}$, Asp$^{217}$, and Asp$^{252}$ form a triad of electrostatic interactions with the Arg$^{193}$ guanidinium at the center forming two hydrogen bonds to the carbonyl groups of Ala$^{254}$ and Asp$^{252}$, (ii) above the Zn$^{2+}$ ion there is a hydrophobic cluster of residues formed by residue Leu$^{147}$, Leu$^{151}$, Leu$^{185}$, Val$^{192}$, and Val$^{195}$ that are located such that they likely form or shape the S1 pocket that accommodates the P1 residue (Tyr$^{1605}$) in VWF (iii) above and below the Zn$^{2+}$ ion two surface-exposed acidic residues Asp$^{187}$ and Glu$^{233}$ are unique to ADAMTS13, (iv) the α-helix in the 231–263 loop of ADAMTS1, 4 and 5 is not formed in ADAMTS13 (Fig. 7c), (v) ADAMTS13 lacks a disulfide bond in the Ca$^{2+}$-binding loop (Ca1) that is present in ADAMTS1, 4, and 5 (Fig. 7c).

The most interesting feature of the ADAMTS13 protease structure is the triad of charged residues formed between Arg$^{193}$, Asp$^{217}$, and Asp$^{252}$. When metalloproteases accommodate their substrates, they do so in a standard orientation with the P1 residue in the substrate engaging with the S1 subsite in the MP domain, positioning the scissile bond over the catalytic center[13]. The P1′ residue is also accommodated by the S1′ pocket. In these proteases, the S1′ pocket is frequently a deep pocket capable of harboring the R-group of the P1′ residue[21]. From the crystal structures of ADAMTS1, 4, and 5 in the presence of active-site inhibitors, the active-site cleft is open and the S1′ pocket is revealed (Fig. 8a–c)[33–35]. In ADAMTS13, the sides of the active-site cleft are formed by the intimate interactions between loop 180–193 and loop 231–263 mediated by side chains from Arg$^{193}$, Asp$^{217}$, and Asp$^{252}$. However, the active-site cleft is occluded by the Ca$^{2+}$-binding loops 180–193 that folds across the cleft and blocks the S1′ pocket (Fig. 8d). The S1′ pocket is shaped by residues Asp$^{252}$-Pro$^{256}$[21]. Therefore, whereas the Zn$^{2+}$ ion is accessible, a triad of salt bridge interactions between the side chains of Arg$^{193}$ with both Asp$^{217}$ (on the α4-helix that runs along the base of the cleft) and Asp$^{252}$ (in the loop 231–263) appear to stabilize the closure of the active-site cleft (Fig. 8e and Supplementary Movie 2). We propose, therefore, that this structure represents a latent conformation that is unable to accommodate a peptide substrate. Arg$^{193}$, Asp$^{217}$, and Asp$^{252}$, herein termed the "gatekeeper triad," blocks both the S1′ pocket and the passage that VWF must take if it is to interact with both

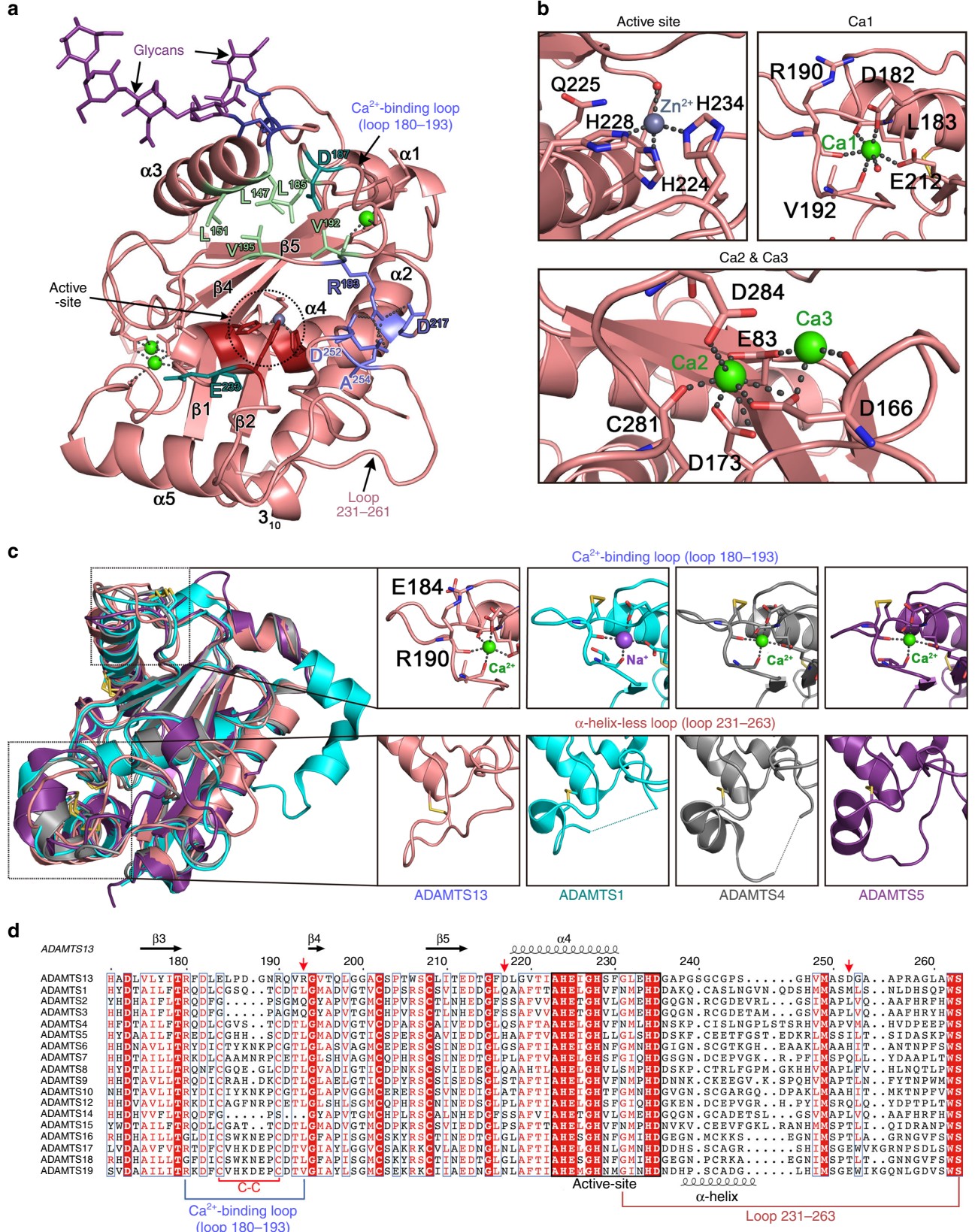

the Dis domain exosite (located at the end of the active-site cleft) and the active-site itself (Supplementary Movie 2). Arg[193], Asp[217], and Asp[252] are unique to ADAMTS13 (Fig. 7d), but are conserved across an amino acid sequence alignment of species from the human to zebrafish ADAMTS13.

## Discussion

Of the 19 ADAMTS family members, ADAMTS13 is the only member with a known role in blood plasma. The importance of its physiological role is highlighted by the severe thrombotic phenotype associated with its inherited or acquired deficiency[5].

**Fig. 7** Structure and sequence comparison of ADAMTS13 metalloprotease (MP) domain. **a** Crystal structure of the ADAMTS13 MP domain with the topology β1-α1-α2-β2-α3-β3-β4-β5-α4-α5. Loop 180–193, loop 231–263, the active-site, and the metal-binding sites are indicated. Two N-linked glycans are shown in purple. Surface residues Leu[147], Leu[151], Leu[185], Val[192], and Val[195] (light green) form a hydrophobic patch above the active-site that likely contributes to the S1 pocket. Asp[187] and Glu[233] (dark teal) are negatively-charged residues that are unique to ADAMTS13. To the right of the active-site, Arg[193], Asp[217], Asp[252] and Ala[254] (blue) form ionic interactions with each other that bridge loops 180–193 and 231–263. **b** Four metal ions are bound by the MP domain: the Zn[2+] ion (gray sphere) is coordinated by His[224], His[228], and His[234]. Three Ca[2+] ions are bound (green spheres); a single Ca[2+] ion (Ca1) bound by loop 180–193 is coordinated by Asp[182], Glu[212] and the carbonyl backbone of Leu[183], Arg[190], and Val[192]. Two Ca[2+] ions (Ca2 and Ca3) are bound in a double Ca[2+]-binding site on the rear side of the MP domain. **c** Superimposition of human ADAMTS13 MP domain (salmon) with the structures of human ADAMTS1 (light blue, 1.19 Å root-mean-square deviation (RMSD), 971 Cα atoms, PDB 3Q2G [https://doi.org/10.2210/pdb3Q2G/pdb]), ADAMTS4 (gray, 1.10 Å RMSD, 923 Cα atoms, PDB 4WK7 [https://doi.org/10.2210/pdb4WK7/pdb]), and ADAMTS5 (purple, 1.10 Å RMSD, 832 Cα atoms, PDB 3LJT [https://doi.org/10.2210/pdb3LJT/pdb]). Regions enclosed with black boxes are enlarged in the right panel. Top panels show the Ca[2+]-binding loop (loop 180-193). ADAMTS13 Glu[184] and Arg[190] interact ionically in place of the disulfide bonds that are found in ADAMTS1, ADAMTS4, and ADAMTS5. Bottom panels highlight differences in the loop 231–263. In ADAMTS1, 4 and 5 in there is an α-helix in this loop, whereas in ADAMTS13 this α-helix is not formed and is presented as the α-helix-less loop. **d** Sequence alignment of ADAMTS13 MP domain residues 171–263 with other ADAMTS family members. Identical residues are colored white on a red background and similar residues are red on white background. The Ca[2+]-binding loop (loop 180–193) is highlighted. "Gatekeeper triad" residues Arg[193], Asp[217], and Asp[252] are denoted by red arrows, which are not present in other ADAMTS family members

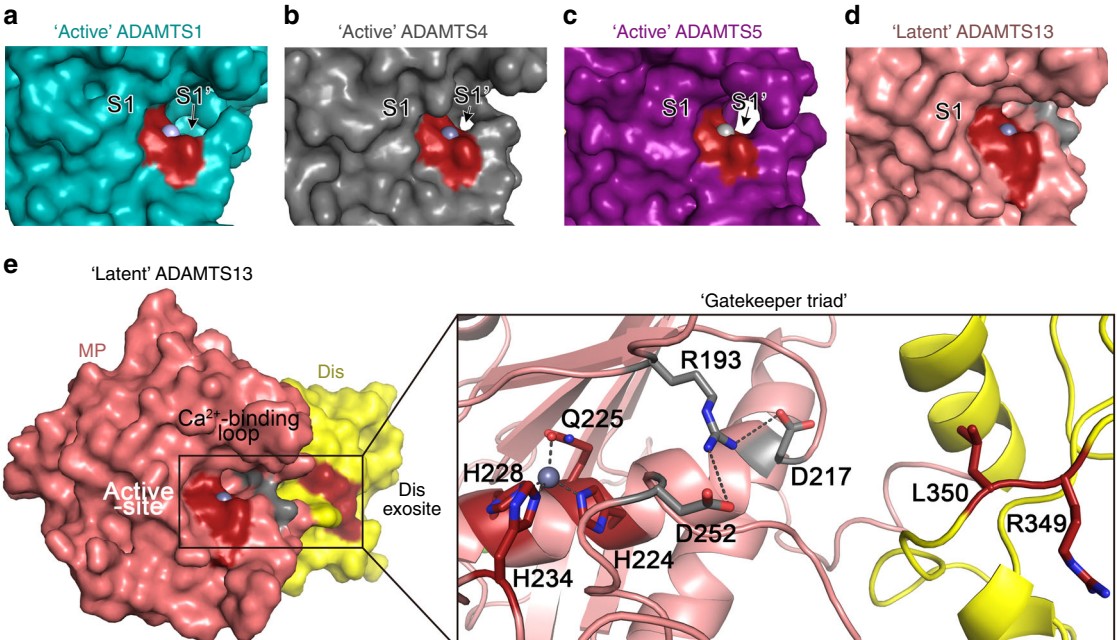

**Fig. 8** The ADAMTS13 metalloprotease (MP) domain exhibits a "closed" conformation and a "gatekeeper triad." **a–c** Structures of the active-sites of ADAMTS1, 4, and 5 resolved in the presence of active-site inhibitors (not shown) revealing the conformation of the "active" protease when the active-site accommodates a substrate PDB ID: 3Q2G [https://doi.org/10.2210/pdb3Q2G/pdb], 4WK7 [https://doi.org/10.2210/pdb4WK7/pdb], and 3LJT [https://doi.org/10.2210/pdb3LJT/pdb], respectively. Active-site His residues are shown in red with bound Zn[2+] ion shown as a gray sphere—all structures are shown in the same orientation. In the standard orientation for MP domain substrate engagement, the scissile bond lies over the active-site. The R-group of the P1 residue is accommodated by the S1 pocket (highlighted). The S1' pocket is recessed into the MP domain to accommodate the P1' residue R-group (arrows). **d** In ADAMTS13, the S1' pocket is occluded due to ionic interactions of the "gatekeeper triad" (gray) residues giving a "latent" conformation. **e** Structure of the ADAMTS13 MP-disintegrin-like (Dis) domains. The active-site and Dis exosite are shown in red, both of which interact with Von Willebrand factor (VWF). The Ca[2+]-binding loop above the active-site cleft forms an ionic interaction via Arg[193] with both Asp[217] and Asp[252] (gray). This ionic interaction of the "gatekeeper triad" blocks the active-site cleft and hence the passage between the active-site and the Dis exosite—see Supplementary Movie 2

Where known, the other family members function primarily in a tissue milieu with substrate targets broadly associated with matrix and cartilage homeostasis[36]. The proteolytic targets of many of the ADAMTS family proteases remains undefined, although a common theme is their specificity for large, often multimeric, structural glycoproteins. All ADAMTS family members contain the MDTCS domains, but differ in the number and identity of C-terminal domains. The conservation of the MDTCS region points to a common mode of action for these domains. Our functional studies reveal that ADAMTS13 contains exosites in each of its

Dis, Cys-rich, and Spacer domains that all contribute to the molecular zipper model of substrate recognition and proteolysis. Whereas the presence of these exosites has been known for some time[15,17–25], their relative importance and biochemical contributions to proteolysis had not been elucidated. Our kinetic data reveal that the Cys-rich and Spacer domain exosites are primarily involved with substrate recognition/binding due to the large increases in both the $K_m$ and $K_D$ associated with disruption of their binding sites in VWF96. The Dis domain also harbors a VWF-binding site primarily involving ionic interactions that,

when it engages VWF, enhances the proteolytic function of the MP domain. This effect was measured through a 56-fold decrease in the $k_{cat}$ (and 14-fold increase in $K_m$) for VWF96 proteolysis when the Dis exosite-binding region in VWF96 was disrupted. For mutations in the substrate to exert such an effect upon the function of the MP domain strongly supports the contention that engagement of this exosite with VWF induces a conformational change to the adjacent MP domain that allosterically enhances its enzymatic function. This substrate-induced activation of the enzyme adds a further level of complexity to the molecular zipper mechanism that describes ADAMTS13 function.

The crystal structure of the MDTCS domains, which is currently the only one to encompass all of these domains of any of the 19 ADAMTS family members, revealed that the $Ca^{2+}$-binding loop 180–193 in the MP domain occludes the active-site cleft and a ring of positive charge defines the substrate-binding interface spanning the MP and Dis domains. This seemingly latent conformation is stabilized by ionic interactions between the "gatekeeper triad" consisting of $Arg^{193}$, $Asp^{217}$, and $Asp^{252}$ that sterically hinders accommodation of a peptide bond across the active-site (Supplementary Movie 2). We propose that this structure corresponds to the circulating latent form of ADAMTS13 and provides important insight into the mode of action for ADAMTS13 (Fig. 9). Normally, VWF circulates in a globular conformation that cannot bind platelets and is resistant to ADAMTS13 proteolysis (Fig. 9a). ADAMTS13 also circulates in a "closed" form mediated by interaction of the CUB domains with the Spacer domain[9,10,37]. The MP domain of circulating ADAMTS13 favors a latent conformation stabilized by the ionic interactions of the "gatekeeper triad." This prevents off-target proteolysis of plasma proteins, and also likely confers resistance to inhibition by broad-spectrum plasma inhibitors as the active-site is not permissive to accommodating a peptide sequence. This would explain how the very long plasma half-life of ADAMTS13 (3.5–8 days) is conferred, which is determined by clearance, as opposed to inhibition[27]. When VWF unravels in response to elevated shear forces (at sites of vessel damage, upon secretion, etc.), it unfolds to facilitate platelet capture. The VWF A2 domain also unfolds to expose the ADAMTS13 cleavage site and exosite-binding regions (Fig. 9b). ADAMTS13 interacts with VWF via the D4-CK region, which "opens" ADAMTS13 up. This enhances ADAMTS13 enzyme activity by ~2-fold against short A2 domain fragment substrates. The Spacer and Cys-rich domains interact with the unfolded A2 domain (Fig. 9c). The Dis domain exosite then engages VWF, which transduces an allosteric change in the MP domain that disrupts the "gatekeeper triad," removing the loop 180–193 from the active-site cleft and converting the latent form into its active conformation (Fig. 9d). This allosteric, substrate-assisted activation enhances ADAMTS13 proteolytic activity of the MP domain by allowing accommodation of the scissile bond into the active-site cleft. ADAMTS13 then dissociates and can recycle to its "closed" and latent form. In this scheme, VWF functions as both the activating cofactor and substrate for ADAMTS13.

The dependence of plasma proteases upon cofactors is a common theme in hemostasis. Activated factor VII is essentially proteolytically inactive unless it is allosterically activated by its cofactor, tissue factor[38]. Activated factor IX is also inert unless bound to its cofactor, factor VIIIa, and is further allosterically activated by binding its substrate, factor X[39]. Similarly, tissue-plasminogen activator is allosterically activated by binding to fibrin[40]. These hemostatic enzymes are all serine proteases, and, to the best of our knowledge, this description of ADAMTS family allostery involving conformational activation of the MP domain may provide mechanistic insight into other family members. Previous descriptions referring to ADAMTS13 allosteric

activation have used this term to describe the opening of ADAMTS13 by dissociation of the CUB domains from the Spacer domain[10–12,37], rather than more classical enzyme allostery associated with structural shifts in the protease domain. We propose that the allosteric activation of ADAMTS13 serves to spatially and temporally localize its proteolytic function and confer its high substrate specificity.

ADAMTS family members have a propeptide that can be removed by furin/furin-like proteases[36]. These propeptides contain a Cys-switch motif that interacts with the active-site, conferring latency until it is removed—either intracellularly or at the cell surface. This inhibits proteolytic activity and prevents proteolysis of intracellular proteins prior to secretion. The ADAMTS13 propeptide is much shorter than those of all other family members and does not possess a Cys-switch motif, and consequently does not confer enzyme latency[7,41]. That the ADAMTS13 MP domain appears to naturally assume a latent conformation may be an evolutionary trait that accommodates its non-inhibitory propeptide. The "gatekeeper triad" residues are unique to ADAMTS13 amongst ADAMTS family members, but are perfectly conserved between species, suggesting that these are functionally/evolutionarily important residues (Fig. 7d). Although the "gatekeeper triad" may be specific to ADAMTS13, the adoption of a latent MP domain fold may not. The crystal structures of the MP domain of ADAMTS1 and 4 have previously been resolved in both the absence and presence of small molecule active-site inhibitors and suggest the need for a structural shift to accommodate their substrates[33,35]. Whether substrate-induced enzyme activation is a common mode of action amongst ADAMTS proteases remains to be determined. However, given the conservation of the MDTCS domains between all members, ADAMTS13 may provide a template for understanding the biochemistry for this family of metalloproteases.

## Methods

**Preparation of VWF A2 substrates**. The coding region for the 96 amino acid A2 domain fragment—termed VWF96, VWF ($Gly^{1573}$-$Arg^{1668}$), fused to an N-terminal 6xHis-SUMO tag (Invitrogen) and a C-terminal HSV-6xHis in pET25b (+) (Novagen) was generated and verified by sequencing. Primer sequences for amplification and cloning into pET-SUMO vector and subsequent subcloning into pET25b(+) are provided in Supplementary Table 2. The 13-kDa N-terminal 6xHis-SUMO tag (Invitrogen) fuses a derivative of the yeast Smt3 protein, which aids in protein solubility. The C-terminal HSV-6xHis tag (Novagen) is a short peptide sequence (SQPELAPEDPEDVEHHHHHH) for which high-affinity antibodies are available. A panel of variants of this vector were generated to ablate each ADAMTS13 exosite-binding region (Fig. 1). Amino acids that were substituted or deleted were identified by previous mutagenesis studies[19,20,24,25,28], with the intention of ablating or severely disrupting each ADAMTS13 exosite binding interaction. To ablate the Spacer domain exosite interaction, residues $Glu^{1660}$-$Arg^{1668}$ were deleted in VWF87($\Delta$Spacer). A variant was also made in which three hydrophobic residues within this region were substituted (L1664T/V1665N/L1666Q) in VWF96-Spacer. For the Cys-rich domain exosite-binding region, I1642Q/W1644Y/A1647S/I1649Q/L1650Q/I1651Q mutations were introduced in VWF96-Cys, as previously reported[24]. The Dis domain exosite-binding region was mutated (D1614A/I1616Q/K1617E/R1618E/D1622T) to make VWF96-Dis based on the study by Kretz et al.[28]. For the MP domain-binding region, we substituted the P3 residue (L1603N) based on the report of Xiang et al.[25] to generate VWF96-MP. We did not substitute the cleavage site, which forms further interactions with the MP domain. Primers used for the generation of each variant are provided in Supplementary Table 2.

BL21 (DE3) *Escherichia coli* (Invitrogen) were transformed with vectors and single colonies inoculated into LB broth containing 100 μg ml$^{-1}$ ampicillin. These were expanded to 2.5 L cultures with shaking at 37 °C. VWF96 variant expression was induced by the addition of 1 mM isopropyl β-D-1-thiogalactopyranoside (IPTG), followed by expression for 5 h. Harvested cells were resuspended in BugBuster® Master Mix (Novagen) containing lysozyme (0.15 mg ml$^{-1}$), benzonase (25 U ml$^{-1}$), and containing protease inhibitors (Sigma). The cell resuspension was incubated at room temperature for 10 min with mixing and centrifuged at 10,000 × *g* for 20 min at 4 °C. Pellets were resuspended in BugBuster® Master Mix, incubated at room temperature for 10 min with shaking, and centrifuged as before. The supernatants were pooled and cleared through

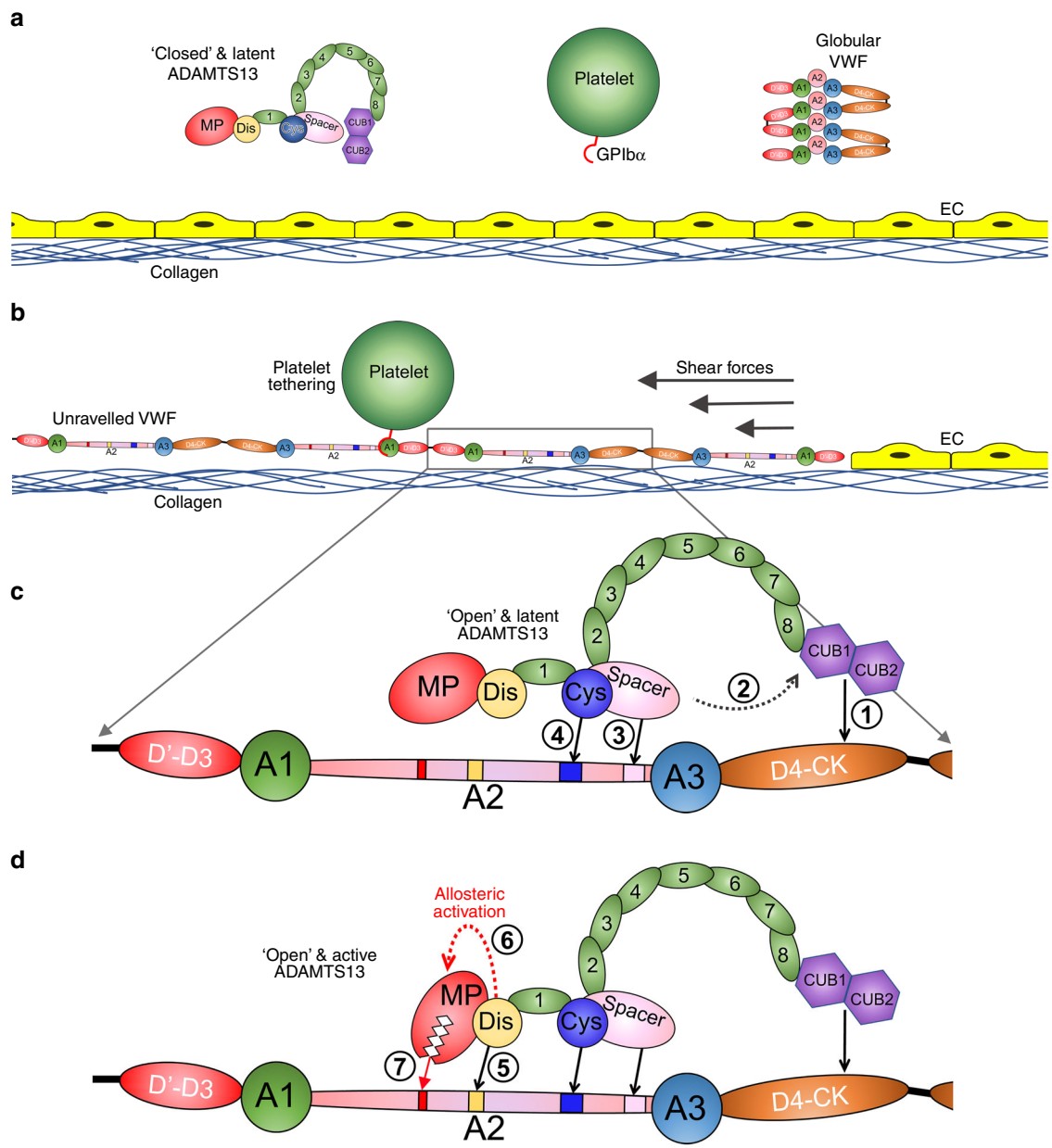

**Fig. 9** Mode of action of ADAMTS13. **a** Under normal circumstances, multimeric Von Willebrand factor (VWF) circulates in the plasma in a globular conformation, in which its A1 domains are concealed, and so does not interact with platelets. ADAMTS13 circulates in a "closed" conformation stabilized through the interaction of the C-terminal CUB domains with the central Spacer domain. The MP domain of ADAMTS13 also has a latent conformation in which the active-site cleft is occluded by the $Ca^{2+}$-binding loop. This prevents ADAMTS13 from proteolyzing off-target substrates and confers resistance to plasma inhibitors. **b** Following vessel damage, the endothelium (EC) is disrupted to reveal subendothelial collagen. Globular VWF binds to this surface via its A3 domain and unravels into an elongated conformation in response to the shear forces exerted by the flowing blood. This reveals the A1 domain that can then capture platelets via the GPIbα receptor on the platelet surface. Unravelling of VWF also unravels the VWF A2 domain into a linear polypeptide conformation that reveals the binding sites for ADAMTS13 and the $Tyr^{1605}$-$Met^{1606}$ cleavage site, making it susceptible to proteolysis by ADAMTS13. **c** ADAMTS13 recognizes unfolded VWF through multiple interactions. (1) The CUB domains bind the VWF D4-CK domains, which (2) induces their dissociation from the Spacer domain. (3) The Spacer and (4) cysteine (Cys)-rich domain exosites recognize the C-terminal region of the unfolded A2 domain to bring the enzyme and substrate into proximity. **d** Once bound, (5) the disintegrin-like (Dis) domain exosite engages VWF residues $Asp^{1614}$-$Asp^{1622}$. This interaction (6) induces an allosteric change in the MP domain. This causes a conformational change, disrupting the "gatekeeper triad" that otherwise occludes the active-site cleft, to reveal the S1' pocket. Once allosterically activated, (7) the MP domain proteolyzes the scissile bond

0.2 μm filters. The filtered lysate was applied to a HiTrap® Chelating HP column (GE Healthcare) bound to $Ni^{2+}$ and equilibrated with 20 mM Tris-HCl (pH 7.6) and 0.5 M NaCl. VWF96 and variants thereof were eluted with step wise imidazole concentrations (30, 60, 100, 150, 200, 250, and 300 mM) in 20 mM Tris-HCl (pH 8.5) and 0.5 M NaCl. Fractions containing VWF96 were identified by sodium dodecyl sulfate-polyacrylamide gel electrophoresis (SDS-PAGE), and dialyzed in 20 mM Tris-HCl (pH 8.5), 50 mM NaCl, and applied to a Capto Q ImpRes column

(GE Healthcare). The column was then washed by 20 mM Tris-HCl (pH 7.6), 50 mM NaCl, and VWF96 variants were eluted from the column by NaCl gradient from 50 mM to 2 M. Purified proteins were analyzed by SDS-PAGE and Coomassie staining, and also by Western blotting using 0.12 μg ml$^{-1}$ anti-SUMO/SUMOStar (LifeSensors; AB7002) and 0.2 μg ml$^{-1}$ anti-HSV (Bethyl; A190-136A) antibodies (Supplementary Fig. 1). Purified proteins >95% pure were dialyzed in 20 mM Tris (pH 7.6), 50 mM NaCl, 5 mM $CaCl_2$ (TBSC), and quantified by absorbance at

280 nm using the calculated extinction coefficients of each variant. Expression vectors for VWF96 and its variants and for samples of recombinant proteins are available upon reasonable request.

**ADAMTS13 activity assays.** For qualitative analysis of VWF96 proteolysis, concentrated recombinant ADAMTS13 in conditioned medium and purified VWF96 variants in TBSC were preincubated at 37 °C for 15 min. Reactions (final volume 135 µl) were set up using 0–53 nM ADAMTS13 and 3 µM VWF96 variants in TBSC buffer. Twenty-five microliters of reaction samples were removed and stopped every 30 min for 1.5 h by mixing with EDTA. Samples were analyzed by SDS-PAGE and Coomassie staining.

For kinetic analyses, ADAMTS13 in conditioned medium and purified VWF96 variants were incubated separately in TBSC/1% bovine serum albumin (BSA) at 37 °C for 15 min, as previously described[20,21,23–26]. Reactions were set up using 0.75–505 nM ADAMTS13 and VWF96 variants (0.5–300 µM) in TBSC/1% BSA. Ten microliters of reaction sub-samples were stopped between 0 and 120 min with EDTA. The stopped sub-samples were diluted to 0.09 nM VWF96 in TBS/1% BSA buffer and analyzed by VWF96 ELISA to quantify the concentration of uncleaved substrate.

**VWF96 ELISA and kinetics of proteolysis.** Polyclonal chicken anti-SUMO/SUMOstar (0.5 µg ml⁻¹) IgY (LifeSensors; AB7002) was adsorbed onto 96-well microtiter plates in 50 mM sodium carbonate/bicarbonate (pH 9.6) overnight at 4 °C. Wells were washed with phosphate-buffered saline (PBS)/0.1% Tween and blocked with TBS/3% BSA for 2 h. Wells were washed, and a standard curve of purified VWF96 (0–0.15 nM) diluted in TBS/1% BSA was added. In parallel, stopped samples were diluted in TBS/1% BSA to a final concentration of 0.09 nM VWF96 and incubated at room temperature for 1.5 h. After washing, 0.5 µg ml⁻¹ peroxidase-conjugated anti-HSV IgG (Bethyl; A190-136P) diluted in TBS/1% BSA was added and incubated for 1 h. Wells were washed and 170 µl SIGMA*FAST*™ *o*-phenylenediamine dihydrochloride peroxidase substrate (Sigma) was used for the detection. Color development was stopped using 2.5 M $H_2SO_4$ and measured spectrophotometrically at 492 nm.

The concentration of uncleaved substrate at each time-point was measured from the standard curve and, from this, the fraction of substrate proteolyzed was calculated and plotted as a function of time. For each VWF96 variant the number of reactions performed is provided in the figure legends. The catalytic efficiency, $k_{cat}/K_m$, was determined using GraphPad software (Prism) by fitting the data from the time-course reactions into the equation, $P = 1 - \exp(-1 \times [\text{ADAMTS13}] \times t \times k_{cat}/K_m)$, where $P$ is the fraction of VWF96 proteolyzed and $t$ is the time in seconds.

To obtain the separate kinetic constants, $k_{cat}$ and $K_m$, for VWF96 variant proteolysis, multiple time-course reactions were performed as described above to determine the initial rate of substrate proteolysis as a function of VWF96 variant concentration (0.2–350 µM). The number of reactions/concentrations performed for each substrate is provided in the figure legends. All assays were performed on multiple different preparations of both recombinant substrates and ADAMTS13. From each time-course analysis, the initial rate of proteolysis (at <15% proteolysis) was calculated (in nM s⁻¹), normalized per unit (nM) of ADAMTS13 used, and plotted as a function of substrate concentration. The Michaelis–Menten equation was used to fit the data using the GraphPad software (Prism), $V_i = (V_{max} \times [\text{VWF96}])/(K_m + [\text{VWF96}])$, where, $V_i$ is the initial rate (nM s⁻¹), $V_{max}$ is the maximum velocity, $K_m$ is the Michaelis constant, and $k_{cat}$ (s⁻¹) is the catalytic constant for proteolysis, or turnover number, which measures the functionality of the active-site of each unit of enzyme ($k_{cat} = V_{max}/[\text{ADAMTS13}]$).

**Binding assay between ADAMTS13 and VWF96 variants.** Binding assays using immobilized VWF96 were performed essentially as previously described[20,21,23–26]. In 96-well microtiter plates, 100 nM VWF96 or VWF96 variants, diluted in 50 mM sodium bicarbonate buffer (pH 9.6), were adsorbed overnight at 4 °C. Wells were washed with PBS/0.1% Tween and blocked with 300 µl PBS/3% BSA for 2 h. Wells were washed, and 100 µl of 0–600 nM ADAMTS13 in PBS/1% BSA/80 mM EDTA were added and incubated at room temperature for 1.5 h with shaking. ADAMTS13 was also added to uncoated, blocked wells, to control for non-specific binding. A buffer-only sample was also included to remove any background absorbance. Bound ADAMTS13 was detected using a biotinylated anti-ADAMTS13 TSP2–4 polyclonal antibody (0.2 µg ml⁻¹), as previously described[24]. All conditions were performed in duplicate and the mean for each well derived. After subtracting any signal from control wells, the data were fitted in the GraphPad software (Prism) using the one-site binding equation: $B = B_{max} \times [\text{ADAMTS13}]/(K_{D(App)} + [\text{ADAMTS13}])$, where $B$ is the percentage specific binding, $B_{max}$ is the maximal binding, and $K_{D(App)}$ is the apparent equilibrium dissociation constant. Data are presented as mean ± SEM of three separate experiments each performed in duplicate.

**Expression and purification of MDTCS.** The complementary DNA encoding the canonical human amino acid sequence of the MDTCS domains (Gly⁷⁸-Pro⁶⁸²) (i.e., without the native signal peptide or propeptide) was cloned into the insect cell expression vector pMT-BiP-PURO. Primers used for amplification and cloning are provided in Supplementary Table 2. This vector fuses the Drosophila BiP secretion

signal to the N terminus of the MP domain to enable entry into the S2 cell secretory pathway. The BiP signal peptide is cleaved off prior to secretion by the cells to provide a native N terminus (Gly⁷⁸). At the C terminus there is a 6xHis tag fused to facilitate purification. The active-site Glu²²⁵ was mutated to Gln (E225Q) by site-directed mutagenesis and verified by sequencing. Primers used for mutagenesis are provided in Supplementary Table 2. This vector was transfected into Drosophila Schneider 2 cells using calcium phosphate, and stably transfected cells were selected using Schneider's Drosophila medium (Lonza) containing 10% (v/v) fetal calf serum and 10 µg ml⁻¹ puromycin (Gibco) for 3–4 weeks. Cells stably expressing MDTCS(E225Q) were expanded to 1 L in Ex-cell™ 420 serum-free medium (Sigma)/10 µg ml⁻¹ puromycin in 2.5-L flasks at 27 °C/120 rpm to a cell density of 2–4 × 10⁶ cells ml⁻¹. To these, copper sulfate (0.5 mM final concentration) was added to induce target protein expression. Cells were cultured for a further 7 days and media harvested by centrifugation and filtration.

Conditioned media were concentrated by tangential flow filtration using a 10 kDa filtration unit (Millipore) and dialyzed into 20 mM HEPES (pH 7.5), 0.5 M NaCl, 5 mM $CaCl_2$, and 10 mM imidazole. The dialyzed medium was applied to a HiTrap® Chelating HP column (GE Healthcare) bound with Ni²⁺. The column was washed and MDTCS(E225Q) eluted through step wise increases in imidazole concentration (30, 60, 100, 150, 200, 250, and 300 mM). Fractions were analyzed by SDS-PAGE and Coomassie staining and the 100 mM imidazole elution fractions containing the highest concentrations of MDTCS(E225Q) were dialyzed into 20 mM HEPES (pH 7.5), 150 mM NaCl, and 5 mM $CaCl_2$, and further purified by gel filtration using a HiPrep® 26/60 Sephacryl S-200HR column (GE Healthcare). The elution fractions containing MDTCS(E225Q) with a C-terminal 6xHis tag were concentrated using Amicon centrifugal filter (10 kDa cut-off) to 4–7 mg/ml. For isolation of a stable complex between MDTCS(E225Q) and the 3H9 Fab fragment, proteins were mixed in a 1:1.2 ratio prior to size exclusion chromatography. Expression vectors for ADAMTS13 and its variants and for samples of recombinant proteins are available upon reasonable request.

**3H9 Fab generation.** Hybridoma cells expressing the inhibitory anti-ADAMTS13 monoclonal antibody, 3H9, were produced in a large scale bio-incubator (Celline CL 350, Integra Biosciences)[30,42]. From conditioned media, the 3H9 antibody was purified using protein A-coupled Sepharose FF (GE Healthcare). Fab fragments of 3H9 were generated proteolytically using papain and subsequently isolated using HiTrap® Q HP anion exchange column (GE Healthcare).

**Crystallization and structure determination.** Crystals of the 3H9 Fab-MDTCS (E225Q) complex were grown using sitting-drop vapor diffusion at 20 °C. Initial crystallization conditions were established using screening kits from Hampton Research (Index) and from Molecular Dimensions (JCGS I, II, III, and IV, MIDAS, ProPlex, and Structure Screen I and II). For the optimal growth of the 3H9 Fab-MDTCS(E225Q) crystals, 1 µl (8 mg ml⁻¹) was mixed with 1 µl of precipitant solution (35% (v/v) pentaerythritol ethoxylate (15/4 EO/OH), 200 mM $CaCl_2$, 100 mM HEPES, pH 6.5) and this was equilibrated against a 1 ml reservoir of the precipitant solution. Crystals were flash frozen in a stream of nitrogen at 100 K and diffraction data were collected at the I24 beamline of the Diamond Light Source, UK. Data were processed and scaled using XDS and the CCP4 suite[43,44]. The crystals belonged to space group $P3_221$ and contained one molecule per asymmetric unit. To determine the structure, molecular replacement was used with the program Phaser within CCP4 suite using the available templates of the ADAMTS13 DTCS domains (PDB: 3GHM [https://doi.org/10.2210/pdb3GHM/pdb]), the MP domain of ADAMTS4 (PDB: 4WK7 [https://doi.org/10.2210/pdb4WK7/pdb]), and a Fab antibody fragment structure (PDB: 2HRP) as search models[45]. Iterative cycles of model building were performed using COOT, followed by refinement with Refmac5 and phenix[46–49]. In the electron density loop regions spanning residues 324–330 (Dis domain) and 450–453, 469–479, 488–489, 498–500 (Cys-rich domain) are poorly defined, suggesting that these regions are flexible. A portion of the data (5%) was set aside before the refinement calculations of $R_{free}$. Solvent molecules were added in the later stages of refinement and final crystallographic statistics are summarized in Supplementary Table 1. Structural alignments and figures were generated using PyMOL (http://www.pymol.org) and UCSF chimera[50].

**Preparation of VWF73.** The coding region for the 73 amino acid human VWF A2 domain fragment (Asp¹⁵⁹⁶-Arg¹⁶⁶⁸) was cloned into pET-SUMO (Invitrogen) containing an N-terminal 6xHis-SUMO tag. This vector was transformed into One Shot BL21 Star (DE3) *E. coli* cells (Invitrogen). Cells were cultured in 200 ml LB media containing kanamycin at 37 °C. One millimolar IPTG was added to induce 6xHis-SUMO-VWF73 protein expression and cultured for 4 h. Cells were harvested and VWF73 purified in the same way as VWF96. Fractions containing pure 6xHis-SUMO-VWF73 were identified by SDS-PAGE. 6xHis-SUMO-VWF73 was incubated with His-tagged SUMO protease-1 (LifeSensors) and 2 mM dithiothreitol at 30 °C for 72 h to remove the His-SUMO tag. Tag-free VWF73 was separated from the cleaved His-SUMO tag and SUMO protease-1 by application to a 1 ml HiTrap® Chelating HP column (GE Healthcare) bound to Ni²⁺. Tag-free VWF73 was dialyzed in 20 mM HEPES (pH 7.5), 150 mM NaCl, 5 mM $CaCl_2$, and concentrated using a 3 kDa cut-off centrifugal filter (Amicon) and quantified by

absorbance at 280 nm using the calculated extinction coefficient of tag-free VWF73.

**Isothermal titration calorimetry**. ITC was performed using a MicroCal PEAQ ITC (Malvern Instruments). All buffers/reagents were degassed prior to use. For this, 280 µl of 27 µM MDTCS(E225Q) in 20 mM HEPES (pH 7.5), 150 mM NaCl, and 5 mM CaCl$_2$ was injected into the sample cell at 25 °C and buffer only into the control cell. Repeated 0.4–2 µl injections of 279 µM VWF73 in 20 mM HEPES (pH 7.5), 150 mM NaCl, and 5 mM CaCl$_2$ were applied to both cells every 150 s with mixing (500 rpm). Thermograms were obtained and fitted via nonlinear least-squares minimization method to determine the $K_D$ and change in enthalpy of binding ($\Delta H$). The Gibbs free energy of binding, $\Delta G$, was calculated from $K_a$ values and the entropic term, T$\Delta S$, was derived from the Gibbs–Helmholtz equation using the recorded $\Delta H$ values. The data presented are the mean ± SEM of four separate experiments.

**Reporting summary**. Further information on research design is available in the Nature Research Reporting Summary linked to this article.

## Data availability

Data supporting the findings of the study are available from the corresponding author upon reasonable request. Protein coordinates and structure factors have been deposited in the RCSB Protein Data Bank under code 6QIG. The source data underlying Figs. 1f, 2a-1, 3a, b and Supplementary Figs. 1a–d are provided as a Source Data file.

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

## Acknowledgements

This work was funded through by a studentship grant from the British Heart Foundation (FS/14/44/30962) awarded to J.T.B.C., a grant from the Medical Research Council (MR/M010260/1) jointly awarded to J.T.B.C. and J.E., and a project grant from the British Heart Foundation jointly awarded (PG/18/17/33572) to J.T.B.C. and J.E. Part of this work was funded by the European Framework Program for Research and Innovation (Horizon2020 Marie Sklodowska Curie Innovative training network PROFILE grant 675746) awarded to K.V.

## Author contributions

A.P. performed the experiments, analyzed the data, prepared the figures, and wrote the manuscript. H.J.K. performed the experiments, analyzed the data, prepared the figures, and wrote the manuscript. Y.X. performed the experiments, analyzed the data, and wrote the manuscript. R.d.G. designed experiments, analyzed the data, and critically reviewed the manuscript. C.L. performed experiments and critically reviewed the manuscript. A.V. performed experiments and provided essential reagents. K.V. provided essential reagents, designed experiments, and critically reviewed the manuscript. J.E. designed experiments, analyzed the data, prepared the figures, and wrote the manuscript. J.T.B.C. designed experiments, analyzed the data, prepared the figures, and wrote the manuscript.

## Additional information

**Competing interests:** The authors declare no competing interests.

