## [Peer Review File · Nature Communications]

Reviewers' comments:

Reviewer #1 (Remarks to the Author):

I had the pleasure to review this manuscript entitled "Crystal structure and substrate-induced activation of ADAMTS13" by Petri et al.

This is a well written and interesting manuscript with detailed biochemical characterization of ADAMTS13 and substrate interaction. The results are quite convincing and potentially useful for our further understanding of the biology of ADAMTS13 and VWF proteolysis. I have a several other comments as the followings.

1) I believe that the first part of manuscript that focuses on biochemical characterization of ADAMTS13 interaction with a small substrate VWF84 or less is somewhat confirmatory of many previous studies. Most of the results are consistent with previously published data including the corresponding author's own group.

2) The interactions between a small substrate VWF84 or less and ADAMTS13 may be quite different from the interactions between ADAMTS13 and a multimeric VWF, particularly with those newly released and anchored on endothelial cells. The newly released ultra large VWF multimers or polymers are surprisingly sensitive to proteolysis without shear, stretch, or flow. Therefore, the results obtained in this study may not reflect the true biology under physiological conditions. These findings are yet to be confirmed in animal models or full-length VWF substrate.

3) The second part of manuscript is about crystal structure of 3H9Fab-MDTCS. This part should stand alone. It does not appear to have much connection with the first part of biochemical study, as the first part results are not used to support the crystallization study, although it is quite interesting. I believe that authors simply combine these two parts to make the manuscript look impressive. Yet, they are independent part of the study.

4) The structure of antibody-MDTCS complex provide some new information regarding the relationship between protease domain and disintegrin domain. However, such a result is incremental from previous DTCS structure. I am not sure that authors have sufficient data to support the hypothesis of substrate induced conformational change in metalloprotease domain. Hydrogen exchange with mass spectrometry technology in the presence or absence of substrate may help strengthen such a hypothesis.

Reviewer #2 (Remarks to the Author):

Platelet-rich thrombi occlude blood vessel that leads thrombosis. Regulation of thrombosis is thus highly important to prevent vessel occlusion. Plasma adhesive large molecular weight protein, von Willebrand factor (vWF), serves as the main player for platelet thrombus formation. Plasma metalloprotease ADAMTS13 regulates vWF platelet-tethering function through cleavage of a single peptide bond in the A2 domain of vWF. A series of previous reports mainly performed by the authors' group indicated continuous multiple site interactions between a peptide chain of the vWF-A2 domain and the proximal domains of ADAMTS13. In addition, the crystal structures of the proximal domains of ADAMTS13, the DTCS domains lacking the metalloprotease (M) domain, has been previously determined.

In the present study, authors evaluated the contribution of each proximal domain for the vWF binding through kinetic parameter (K_m and k_{cat}) determination and then determined the crystal structure of MDTCS complexed with an antibody fragment. Based on these experimental pieces of evidence, authors disclosed a novel and sound mechanism how ADAMTS protease recognizes the specific substrate and finally allosterically changes the conformation to enable accommodation of the substrate. Crystal structure of the M domain revealed a closed form of the active site cleft, indicating a latent conformation unable to accommodate a peptide substrate. The substrate peptide binding to the disintegrin (D) domain induces conformational changes in the M domain that leads to the open conformation to accommodate a peptide substrate. The scenario of ADAMTS13 activation revealed in the present study can be applicable to other ADAMTS proteases

that are involved in many biological processes. The present study gives a novel insight into the regulation of ADAMTS proteases, that leads to new therapeutics.

Minor comments:

Page 3, line 17, page 6, line 21: Ref 9 is not a crystal structure paper.

Page 6, line 2: Authors discussed that the VWF96 fragment would be in equilibrium between open and closed states. However, the NMR analysis of ¹H and ¹⁵N double-labelled VWF73-H and VWF64-H indicated an extended structure for both peptides, suggesting an induced-fit substrate recognition mechanism (Sadler et al, Hematology Am Soc Hematol Educ Program. 2004:407-23).
Page 6, Crystal structure of Fab-MDTCS complex: MDTCS has glycosylation sites. Did MDTCS expressed with insect cells have an O-linked sugar chain at Ser399? Did you see the O-linked sugar chain at Ser399 in the structure?

Page 11, line 4: The VWF96 fragment and its derivatives have a His-SUMO tag and an HSV-His tag. I ask to describe the amino acid sequences of the SUMO and HSV tags in the text. Because amino acid sequences are important information to exclude the possibility that these tags may cause two-state equilibrium of the VWF96 fragment.

Page 13: Expression of MDTCS(E225Q): Description of expression system of MDTCS(E225Q) protein was confusing for me, and I request to rewrite. The insect cell expression vector seems to have the signal peptide and C-terminal tag sequence, but there was no description on them. Also, I would like to know the deduced amino acid sequence the final purified MDTCS(E225Q) protein that was subjected to the crystallization. It seems that crystallized MDTCS(E225Q) protein has tag sequences.

Reviewer #3 (Remarks to the Author):

This study focusses on ADAMTS13, one member of an extensive and important family of proteases referred to as ADAMs. A crucial question awaiting an answer is how ADAMTS13 - and other members of the family - achieves exquisite cleavage specificity despite the fact that the protease domain itself is rather unspecific, and how ADAMTS13-proteolytic activity is switched on when needed.

Although this study does not provide the final and complete mechanism of activation it does present an exciting and important conceptual advance.

The crucial insight obtained from the crystallographic part of this study is the discovery that a loop of the protease domain occupies the active-site cleft preventing substrate binding. This loop is spatially close to a substrate binding exosite in the Dis-domain suggesting that substrate-binding to Dis induces a conformational change that displaces the loop and thereby activates the protease domain.

Besides structural data the manuscript presents supporting enzymatic data, binding and ITC experiments. The crystallography and other experiments appear technically sound. Not all enzymatic data are new (prior data are properly referenced), however, this study does provide the most extensive functional analysis of mutants thus far.

I have some concerns about the work, those regarding the conclusions from the ITC analysis being most serious.

Main points:

1) I totally disagree with the phrasing of the purpose of, and the conclusions drawn from, the ITC experiments. The authors state that they use ITC in this study "to identify potential conformational changes induced by (this) binding". Moreover, when interpreting the results from ITC the authors

state that the unfavorable entropy change of binding is indicative of binding-induced conformational changes and that "This may correspond to allosteric changes in the ADAMTS13 MP domain...". Firstly, from simple ITC measurements performed at one temperature no information on the occurrence of conformational changes can be obtained. The paper referred to (ref 31) discusses how ITC-derived temperature-dependent changes in heat-capacity $\Delta C_p(T)$, obtained by performing ITC measurements at different temperatures, enables discrimination between induced fit and conformational selection mechanisms of binding. This analysis does not apply to the current study. In my opinion the authors cannot draw a conclusion on the occurrence of a conformational change from their ITC data.

2) The authors state they use Fab 3H9 to stabilize the proteolytic domain. This raises the obvious question whether the conformation of the domain could have been affected by Fab binding. The authors should address this point in the main text.

3) Although performed in a similar fashion in several other published studies about the enzymatic function of ADAMTS13, it eludes me what the added value is of determining k_{cat}/K_m values in two different ways. First from a time course experiment and next from initial reaction rates at different substrate concentrations. The Michaelis-Menten equation dictates measurement of initial reaction rates (so in the linear range of the reaction) or otherwise the approximations used in their derivation are invalid. Determination of k_{cat}/K_m values from a single time course experiment is not very reliable, partly so because effects of product inhibition may start to play a role at later time-points. Why use this poor man's approach if the accurate approach can be, and in fact is, used?

Minor remarks:

Introduction: "VWF plasma concentration and multimer size are risk factors for thrombosis:"
Should probably read "High VWF plasma concentration and large multimer size ..."

Introduction: The multiple exosite interactions have given rise to the so-called "molecular zipper" mechanism of ADAMTS13. Though evolution may have given rise to this particular mechanism, in the current context what is probably meant is that exosite interactions have given rise to the so-called "molecular zipper" model of ADAMTS13 function.

Technically, the proteolytic domain does not bind VWF-A2 residue Leu1603 at an exosite. The P3 pocket that accommodates this residue is in the active center and can therefore not be considered an exosite. Description of all mutated sites as exosites is convenient and I guess acceptable, but this exception should be pointed out.

I am confused by the sentence "Kinetic analyses demonstrated that, in binding VWF, the ADAMTS13 cysteine rich and spacer-domain exosites approximate the enzyme and substrate". The Cambridge dictionary does not describe the use of approximate in this context. I guess it should mean approach. If so, how can such a conclusion be drawn from kinetic data?

Figure 1d. Why not move the text "VWF73 6xHis-SUMO" in the last line to the left and indicate the absence of 1573-1595 by a line as for VWF87(Δ spacer)?

Legend to Figure 1f. Is the usage of "VWF115" correct here? From the gel its molecular weight appears to be lower than that of VWF96. Also the term VWF VWF115 is nowhere else used in the manuscript. Does figure 1f perhaps show VWF73?

"Equilibrium plate binding assay" a google search on this phrase indicates that it is only used in ADAMTS13 literature and by a limited set of authors. The term should not be used; it suggest that the method employed measures equilibrium binding, which it does not, because it is sensitive to the off-rate.

Several values mentioned in the text about "Equilibrium plate binding assay" appear to be inaccurate. For instance the increase in $K_D(\text{app})$ for VWF96-Cys is 16-fold not 15-fold, whereas the increase for VWF-Dis binding is 15-fold not 14-fold.

Page 7 half way and legend of Figure 6: "The electrostatic charged surface..." is incorrectly phrased. What is displayed in figure 6 is the electrostatic potential at the molecular surface.

Page 8: In the enumeration of differences between ADAMTS13 and ADAMTS1, 4 and 5. Point (iv) the alpha-helix is deleted probably should read "the alpha-helix is not formed", because the sequence alignment in Fig 7 suggests that the residues that form the helix are present in ADAMTS13.

Reviewers' comments:

Reviewer #1 (Remarks to the Author):

I had the pleasure to review this manuscript entitled "Crystal structure and substrate-induced activation of ADAMTS₁₃" by Petri et al. This is a well written and interesting manuscript with detailed biochemical characterization of ADAMTS₁₃ and substrate interaction. The results are quite convincing and potentially useful for our further understanding of the biology of ADAMTS₁₃ and VWF proteolysis.

We thank the reviewer for his/her kind comments regarding our study. We too believe in the impact of this work towards the understanding of the ADAMTS₁₃ and VWF field, but also more widely to the ADAMTS and metalloprotease fields as well.

I have a several other comments as the followings.

1) I believe that the first part of manuscript that focuses on biochemical characterization of ADAMTS₁₃ interaction with a small substrate VWF₈₄ or less is somewhat confirmatory of many previous studies. Most of the results are consistent with previously published data including the corresponding author's own group.

The reviewer is correct that we have previously explored the contribution of different exosites in ADAMTS₁₃ and the reciprocal binding sites in VWF. However, this is the first time this substrate (VWF₉₆) has been used and that the contribution of each exosite interaction has been probed in parallel. The results for VWF₉₆ and its derivatives are certainly confirmatory in part, as we reference. That said, our data are the first to derive the individual *k*_{cat} and *K*_m parameters for each of the exosite disruptions, and also to do them in parallel. The most important and insightful data are derived from the disruption of the Dis domain exosite interaction. This is the first time that disruption of the entire Dis site has been analysed (rather than substitution of just individual amino acids), and the first time that individual constants *k*_{cat} and *K*_m have been derived for this. The strength of these kinetic data are, at least in part, linked to the parallel analysis of the different exosite disruptions demonstrating how robust the assays are and to thoroughly detail their relative contributions/importance.

2) The interactions between a small substrate VWF₈₄ or less and ADAMTS₁₃ may be quite different from the interactions between ADAMTS₁₃ and a multimeric VWF, particularly with those newly released and anchored on endothelial cells. The newly released ultra large VWF multimers or polymers are surprisingly sensitive to proteolysis without shear, stretch, or flow. Therefore, the results obtained in this study may not reflect the true biology under physiological conditions. These findings are yet to be confirmed in animal models or full-length VWF substrate.

We agree that the interaction between full length ADAMTS₁₃ and full length multimeric VWF is more complex than with truncated A₂ domain fragment substrates in part due to the role of the C-terminal tail of ADAMTS₁₃ and other domain interactions with full length VWF. However, it is clear that the N-terminal domains of ADAMTS₁₃ (MDTCS) are involved specifically with the interaction and proteolysis of the unfolded VWF A₂ domain. The use of shorter VWF A₂ domain fragments to monitor proteolysis has major advantages over the use of full length VWF, largely due to the ability to perform truly kinetic studies to derive kinetic constants, but also to more specifically probe certain interactions. This is simply not possible for multimeric VWF due to the complexity of the quantitation of proteolysis and the need for shear (or denaturants in the reaction) to facilitate cleavage. Confirming the data using full length VWF is also highly problematic due to the contribution of A₂ domain folding to the expression and proteolytic susceptibility of VWF. When introducing A₂ domain substitutions into full length VWF, the potential to disrupt the domain folding is high. There are several examples of amino acids substitutions of residues important for ADAMTS₁₃ binding that cause type 2A VWD (i.e. increased rate of proteolysis of full length VWF in vivo). This is because, although an amino acid may be important for ADAMTS₁₃ recognition, the overriding effect upon VWF is the increased propensity for the A₂ domain to unfold, which actually leads to enhanced proteolysis in vivo. We agree that probing ADAMTS₁₃ function with full length VWF would be desirable, but currently a means/system to do this in a quantitative and meaningful way is lacking.

3) The second part of manuscript is about crystal structure of 3HgFab-MDTCS. This part should stand alone. It does not appear to have much connection with the first part of biochemical study, as the first part results are not used to support the crystallization study, although it is quite interesting. I believe that authors simply combine these two parts to make the manuscript look impressive. Yet, they are independent part of the study.

Broadly speaking, we agree that the two parts of our manuscript are distinct. The first part involves kinetic analyses of proteolysis, whereas the second part involves structural analysis. However, we respectfully disagree

that they are independent. Indeed, the data are mostly presented chronologically (i.e. in the order that the data became available). When we derived the kinetic constants for proteolysis of VWF₉₆ and its derivatives, we hypothesised that the data were consistent with an allosteric/substrate-induced conformational change that might facilitate the proteolytic event. However, direct evidence for this was lacking and was only insinuated. When the crystal structure data became available, this provided a mechanism as to how this conformational change might be manifest given the proximity of the Dis exosite to the location of the loop that must be removed from the active site cleft. The crystal structure now defines for the first time in entirety the arrangement of the "molecular-zipper" of ADAMTS₁₃ exosites and their relative disposition with respect to the active site so we feel this is connected to the biochemical study. For these reasons, we feel that the data are highly complementary and sit perfectly side-by-side.

4) The structure of antibody-MDTCS complex provide some new information regarding the relationship between protease domain and disintegrin domain. However, such a result is incremental from previous DTCS structure. I am not sure that authors have sufficient data to support the hypothesis of substrate induced conformational change in metalloprotease domain. Hydrogen exchange with mass spectrometry technology in the presence or absence of substrate may help strengthen such a hypothesis.

The structure of the ADAMTS₁₃ DTCS domains provided a highly important platform from which to understand ADAMTS₁₃ function. However, those data naturally lacked the structure of the metalloprotease domain that we now present and which lies at the heart of the enzyme's function. We show in our manuscript how the presence of the MP domain alters the conformation/relative positioning of the DTCS domains (when compared to the previous DTCS structure), and also how the presence of the MP domain influences the structure of the Dis domain with which it shares an interface and also reveal, what is clearly a conformation of the MP domain that is unable to accommodate a peptide bond across its active centre. This, in conjunction, with the kinetic data presents compelling evidence for a conformational change/need for a conformational change to enable a proteolytic event. We have naturally been looking for further data sets to corroborate this. Hydrogen-Deuterium exchange (HDX) is a nice idea and something that we are working on. Another is endeavouring to resolve the crystal structure of the complex between MDTCS and VWF₇₃. The biggest issue with these two approaches is that MDTCS and VWF₇₃ do not form a stable complex together. We have tried on multiple occasions to form a stable complex in solution, but to no avail. When gel filtered MDTCS and VWF₇₃ resolve as separate peaks. We now believe we understand the reason for this and are endeavouring to address this issue, but this remains something that is far from being solved. For HDX to work well, one ideally needs a tight complex. Moreover, a linear polypeptide chain such as VWF₇₃ may not provide much 'protection' from HDX to assist in identifying the points of contact between ADAMTS₁₃ and VWF₇₃, let alone those regions that are likely to change conformationally – this represents a challenge that we are working on, but that represents an area for further research

Reviewer #2 (Remarks to the Author):

Platelet-rich thrombi occlude blood vessel that leads thrombosis. Regulation of thrombosis is thus highly important to prevent vessel occlusion. Plasma adhesive large molecular weight protein, von Willebrand factor (vWF), serves as the main player for platelet thrombus formation. Plasma metalloprotease ADAMTS₁₃ regulates vWF platelet-tethering function through cleavage of a single peptide bond in the A₂ domain of vWF. A series of previous reports mainly performed by the authors' group indicated continuous multiple site interactions between a peptide chain of the vWF-A₂ domain and the proximal domains of ADAMTS₁₃. In addition, the crystal structures of the proximal domains of ADAMTS₁₃, the DTCS domains lacking the metalloprotease (M) domain, has been previously determined.

In the present study, authors evaluated the contribution of each proximal domain for the vWF binding through kinetic parameter (K_m and k_{cat}) determination and then determined the crystal structure of MDTCS complexed with an antibody fragment. Based on these experimental pieces of evidence, authors disclosed a novel and sound mechanism how ADAMTS protease recognizes the specific substrate and finally allosterically changes the conformation to enable accommodation of the substrate. Crystal structure of the M domain revealed a closed form of the active site cleft, indicating a latent conformation unable to accommodate a peptide substrate. The substrate peptide binding to the disintegrin (D) domain induces conformational changes in the M domain that leads to the open conformation to accommodate a peptide substrate. The scenario of ADAMTS₁₃ activation revealed in the present study can be applicable to other ADAMTS proteases that are involved in many biological processes. The present study gives a novel insight into the regulation of ADAMTS proteases, that leads to new therapeutics.

We thank the reviewer for his/her positive comments

Minor comments:

Page 3, line 17, page 6, line 21: Ref 9 is not a crystal structure paper.

We apologise for this; this is indeed an issue with incorrect referencing and we thank the reviewer for spotting this. This reference has been removed from the inappropriate locations and has now been corrected on both **pages 3 and 6**.

Page 6, line 2: Authors discussed that the VWF96 fragment would be in equilibrium between open and closed states. However, the NMR analysis of ¹H and ¹⁵N double-labelled VWF73-H and VWF64-H indicated an extended structure for both peptides, suggesting an induced-fit substrate recognition mechanism (Sadler et al, Hematology Am Soc Hematol Educ Program. 2004:407-23).

We are aware of this manuscript. However, as that was a review article and the original data were not included, or indeed reported/published subsequently it has not been possible to interrogate these data appropriately. Indeed, based on our hypotheses, we have ourselves also performed NMR with VWF73 from which we have seen evidence for some structural characteristics that are providing clues as to how we may be able to disrupt this and thus enable a VWF A2 domain peptide to adopt a more open/extended conformation (akin to when it is immobilised on a plate). In our opinion the complexity of the enzyme substrate recognition system employed by ADAMTS₁₃-VWF involving multiple exosites and conformational changes in the substrate and enzyme means it is an oversimplification to use terms such as induced-fit, so we have purposefully avoided this.

Page 6, Crystal structure of Fab-MDTCS complex: MDTCS has glycosylation sites. Did MDTCS expressed with insect cells have an O-linked sugar chain at Ser399? Did you see the O-linked sugar chain at Ser399 in the structure?

We do resolve an O-linked glycan on Ser399. N-linked glycosylation in insect cells is akin to that in mammalian cells (although with shorter more uniform chains). Study of O-link glycosylation in S2 cells is limited and there do appear to be some differences in the nature of O-linked glycosylation. However, Ser399 appears to be O-linked glycosylated as it is in humans. We have updated the text on **page 6** to include this information.

"We detected N-linked glycans on all previously identified sites within MDTCS (Asn142 and Asn146 in the MP domain, Asn552 in the Cys-rich domain, and Asn579, Asn614 and Asn667 in the Spacer domain). In addition, we also resolved an O-linked glycan on Ser399, but not on any of the other previously reported sites."

Page 11, line 4: The VWF96 fragment and its derivatives have a His-SUMO tag and an HSV-His tag. I ask to describe the amino acid sequences of the SUMO and HSV tags in the text. Because amino acid sequences are important information to exclude the possibility that these tags may cause two-state equilibrium of the VWF96 fragment.

We have provided additional information on the tags attached to the VWF fragments within the materials and methods. The VWF96 fragments and its derivatives all have an N-terminal tag consisting of a 6xHis tag followed by the SUMO fusion protein (13kDa) this is flanked with a SUMO protease cleavage site. At the C-terminus there is an HSV tag followed by another 6x His tag. We used VWF115 to compare the rate of proteolysis with VWF96. VWF115 does not have a SUMO or HSV tag (it has a short N-terminal Xpress epitope tag). This suggests that there are no artifactual effects on proteolysis associated with the use of the SUMO and HSV tags.

On **page 11**, we now include

"The 13kDa N-terminal 6xHis-SUMO tag (Invitrogen) fuses a derivative of the yeast Smt3 protein, which aids in protein solubility. The C-terminal HSV-6xHis tag (Novagen) is a short peptide sequence (SQPELAPEDPEDVEHHHHH) for which high affinity antibodies are available."

Page 13: Expression of MDTCS(E225Q): Description of expression system of MDTCS(E225Q) protein was confusing for me, and I request to rewrite. The insect cell expression vector seems to have the signal peptide and C-terminal tag sequence, but there was no description on them. Also, I would like to know the deduced amino acid sequence the final purified MDTCS(E225Q) protein that was subjected to the crystallization. It seems that crystallized MDTCS(E225Q) protein has tag sequences.

We have rewritten the expression of MDTCS to include further details of the vector and the tag. The crystallised material has just a C-terminal 6xHis tag. The BiP signal peptide for the insect expression system is there to enable secretion from the S2 cells, but is cleaved off by the S2 cell machinery at the point of secretion so is not present in the purified material. On **page 13**, we have added

"The cDNA encoding the canonical human amino acid sequence of the MDTCS domains (Gly78-Pro682) (i.e. without the native signal peptide or propeptide) was cloned into the insect cell expression vector pMT-Bip-PURO. This vector fuses the Drosophila BiP secretion signal to the N-terminus of the MP domain to enable entry into the S2 cell secretory pathway. The BiP signal peptide is cleaved off prior to secretion by the cells to provide a native N-terminus (Gly78). At the C-terminus there is a 6xHis tag fused to facilitate purification."

Reviewer #3 (Remarks to the Author):

This study focusses on ADAMTS₁₃, one member of an extensive and important family of proteases referred to as ADAMs. A crucial question awaiting an answer is how ADAMTS₁₃ - and other members of the family - achieves exquisite cleavage specificity despite the fact that the protease domain itself is rather unspecific, and how ADAMTS₁₃-proteolytic activity is switched on when needed.

Although this study does not provide the final and complete mechanism of activation it does present an exciting and important conceptual advance.

The crucial insight obtained from the crystallographic part of this study is the discovery that a loop of the protease domain occupies the active-site cleft preventing substrate binding. This loop is spatially close to a substrate binding exosite in the Dis-domain suggesting that substrate-binding to Dis induces a conformational change that displaces the loop and thereby activates the protease domain.

Besides structural data the manuscript presents supporting enzymatic data, binding and ITC experiments. The crystallography and other experiments appear technically sound. Not all enzymatic data are new (prior data are properly referenced), however, this study does provide the most extensive functional analysis of mutants thus far.

I have some concerns about the work, those regarding the conclusions from the ITC analysis being most serious. **We thank the reviewer for his/her recognition of the conceptual advances provided by our data**

Main points:

1) I totally disagree with the phrasing of the purpose of, and the conclusions drawn from, the ITC experiments. The authors state that they use ITC in this study "to identify potential conformational changes induced by (this binding)". Moreover, when interpreting the results from ITC the authors state that the unfavorable entropy change of binding is indicative of binding-induced conformational changes and that "This may correspond to allosteric changes in the ADAMTS₁₃ MP domain...". Firstly, from simple ITC measurements performed at one temperature no information on the occurrence of conformational changes can be obtained. The paper referred to (ref 31) discusses how ITC-derived temperature-dependent changes in heat-capacity $\Delta C_p(T)$, obtained by performing ITC measurements at different temperatures, enables discrimination between induced fit and conformational selection mechanisms of binding. This analysis does not apply to the current study. In my opinion the authors cannot draw a conclusion on the occurrence of a conformational change from their ITC data.

The ITC data is of high quality and provides an important solution based, label free, measure of the solution binding K_D to provide a rigorous examination of the interaction between ADAMTS₁₃ and VWF. This is the first time the thermodynamics of the ADAMTS₁₃-VWF A2 interaction has been reported. The referee makes a valid point that a conclusion of conformational change cannot be drawn from single temperature experiments (with n=4) and we have altered the wording appropriately and removed the reference. The ITC experiments use very high quantities of recombinant proteins (appreciably more than that required for the crystal structure), which has precluded repetitions of the experiments at different temperatures. On page 6, we have changed the text to read;

"To analyze solution-phase binding between ADAMTS₁₃ and the VWF A2 domain fragment, we employed isothermal titration calorimetry (ITC) using inactive ADAMTS₁₃ MDTCS(E225Q) and tag-free VWF73 (Fig 1). From the isotherms, we derived a solution phase K_D of 450nM, which is closer to the K_m for proteolysis than the $K_{D(App)}$ derived from the plate binding assays (Fig 3b). The isothermal profile for the binding of VWF73 to MDTCS(E225Q) revealed a favorable enthalpic component but unfavorable entropic contribution (Fig 3c)"

We have also removed the statement about a conformational change from the legend of Fig 3 and also from the discussion

2) The authors state they use Fab 3Hg to stabilize the proteolytic domain. This raises the obvious question whether the conformation of the domain could have been affected by Fab binding. The authors should address this point in the main text.

This is an interesting question and supplementary Figure 3 details the Fab:ADAMTS₁₃ interactions. The Fab CDR3 forms several contacts the ADAMTS₁₃ residues 187-189 which are in a β -hairpin loop structure (Figure 7c)

and ADAMTS₁₃ Asp187 contacts the Fab via a salt bridge with residue Arg99. This β -hairpin loop conformation is also observed in the ADAMTS₁ and 4 protease structure (pdb codes 2V4B, 3q2g, 4wk7) where it is maintained with an identical internal main chain hydrogen pattern so it is unlikely that the Fab is affecting the main chain structure in this region but it may affect the position of the Asp187 and Asn198 sidechain rotamers. The Fab CDR1 contacts ADAMTS₁₃ residues Thr148 and Gln197 which both maintain the approximate same relative disposition and main chain conformation/hydrogen bonds in an alpha-helix and beta-hairpin respectively when compared with the structures of related ADAMTS_{1,4,5} proteases. Supp Figure 3d shows CDR2 contacts ADAMTS₁₃ residue Glu233 with Fab main chain and sidechain contacts and this residue is directly N-terminal to His234 which coordinates the zinc ion. Again the main chain of this residue is familiar to the other ADAMTS structures available. Fab residues Tyr59 and Tyr57 contact residues P238 and Q269 respectively. This loop is variable in structure and sequence compared to the other metalloproteases (Figure 7c,d) so it is not possible to determine whether its conformation is affected by the Fab binding.

Overall the ADAMTS₁₃ protease is stabilised by multiple disulphide bonds, bound zinc and calcium ions in the area where the Fab 3H9 binds that maintain the integrity of the local loop main chain structure as observed in other ADAMTS protease structures but local changes in sidechain rotamers will be caused by contact with the Fab. We therefore agree that it remains conceivable that the 3H9 Fab may influence the conformation of the ADAMTS₁₃ MP domain. With this point in mind, we have already endeavoured to obtain structures of MDTCS with other anti-MP domain Fabs, but to no avail. It appears that the location of binding of the 3H9 Fab constrains an otherwise flexible region in the ADAMTS₁₃ MP domain that without constraint prevents crystallisation. We have modified the text to include discussion of the possibility of Fab-induced conformational changes in the results section.

On page 6, we now include

"The overall structure of the ADAMTS₁₃ MP domain is stabilized by multiple disulfide bonds and bound Zn²⁺ and Ca²⁺ ions making it unlikely the inhibitory Fab binding alters the local main chain conformation significantly. Moreover, the conformation of the ADAMTS₁₃ MP when compared to other ADAMTS family MP domains (discussed below) reveals that there are no major differences in the overall structure, however, it is possible Fab binding induces subtle changes the positioning of the flexible loops or local changes in the sidechain rotamers."

3) Although performed in a similar fashion in several other published studies about the enzymatic function of ADAMTS₁₃, it eludes me what the added value is of determining kcat/Km values in two different ways. First from a time course experiment and next from initial reaction rates at different substrate concentrations. The Michaelis-Menten equation dictates measurement of initial reaction rates (so in the linear range of the reaction) or otherwise the approximations used in their derivation are invalid. Determination of kcat/Km values from a single time course experiment is not very reliable, partly so because effects of product inhibition may start to play a role at later time-points. Why use this poor man's approach if the accurate approach can be, and in fact is, used?

We agree that the derivation of the kcat/Km from time course studies is indeed a 'poor man's' approach. However, that approach is particularly useful in screening substrates/variants to determine functional deficiencies in proteolysis prior to formal determination of constants. It also allows one to accumulate data much more rapidly before going through the laborious process of performing the initial rate of proteolysis reactions (derivation of these data took nearly 2 years). Given the excellent agreement in the kcat/Km, we felt that this 'double' approach added strength to the data. The reviewer raises an excellent point about the potential for cleavage product-mediated inhibition. A very interesting finding that we have noted is that the C-terminal cleavage product of VWF96 appears to have very little/no inhibitory effect, even at high molar excess. The reasons for this are potentially linked to the conformation that the cleavage product adopts causing it to not favour an interaction with ADAMTS₁₃. This is something that we are exploring, but that is beyond the scope of the current submission.

Minor remarks:

Introduction: "VWF plasma concentration and multimer size are risk factors for thrombosis:"

Should probably read "High VWF plasma concentration and large multimer size ..."

Yes, we agree with this and have introduced this correction on page 3

Introduction: The multiple exosite interactions have given rise to the so-called “molecular zipper” mechanism of ADAMTS₁₃. Though evolution may have given rise to this particular mechanism, in the current context what is probably meant is that exosite interactions have given rise to the so-called “molecular zipper” model of ADAMTS₁₃ function.

Yes, we agree with this and have introduced this correction on page 3

Technically, the proteolytic domain does not bind VWF-A2 residue Leu1603 at an exosite. The P₃ pocket that accommodates this residue is in the active center and can therefore not be considered an exosite. Description of all mutated sites as exosites is convenient and I guess acceptable, but this exception should be pointed out.

Yes, we agree with this that the S₃ pocket that accommodates the P₃ residue can be considered a part of the active centre. The distinction that we intended to make was to discriminate between the cleavage site specificity/accommodation versus interaction sites outside of these amino acids. We accept this point and have clarified this in the text so that the term exosite is not misleading to the reader. On page 4, we now include ***“Although not strictly an exosite interaction, we also substituted the P₃ residue (L1603N) in VWF96, which is important for recognition by the MP domain S₃ subsite, to evaluate the contribution of MP domain recognition beyond the cleavage site residues.”***

I am confused by the sentence “Kinetic analyses demonstrated that, in binding VWF, the ADAMTS₁₃ cysteine rich and spacer-domain exosites approximate the enzyme and substrate”. The Cambridge dictionary does not describe the use of approximate in this context. I guess it should mean approach. If so, how can such a conclusion be drawn from kinetic data?

We used the term ‘approximate’ to mean bring the enzyme (MP domain) and substrate (cleavage site) into proximity with each other. We draw this conclusion because disruption of these exosite interactions alter the K_m for proteolysis demonstrating that these are binding exosites. Consulting the Cambridge English dictionary, the reviewer is perfectly correct that the meaning that we intend is not well referred to in that dictionary. However, in the Oxford English dictionary (often considered the definitive version) the definition is given as “To bring close or near, to cause to approach or be near (to). Rarely, and chiefly in scientific language, of physical motion (as of molecules), but common in other relations”. However, to avoid confusion, and so as not to cause umbrage between the learned institutions of Oxford and Cambridge, we have modified the text in the **abstract** and in the **legend to Fig 9!**

Figure 1d. Why not move the text “VWF73 6xHis-SUMO” in the last line to the left and indicate the absence of 1573-1595 by a line as for VWF87(Δspacer)?

This is a reasonable suggestion which we have introduced into an updated Fig 1.

Legend to Figure 1f. Is the usage of “VWF115” correct here? From the gel its molecular weight appears to be lower than that of VWF96. Also the term VWF VWF115 is nowhere else used in the manuscript. Does figure 1f perhaps show VWF73?

We apologise for the confusion here. VWF115 is a longer VWF A2 domain fragment than VWF96. However, it does not contain the 13kDa N-terminal SUMO tag or short C-terminal HSV Tag (VWF115 does contain a short 3.2kDa N-terminal Xpress epitope tag). Therefore, the molecular weight does not compare with VWF96 for this reason. We appreciate the confusion associated with this and so have clarified the amino acid regions in each fragment, as well as details about the MW of the tags on these in the text and in the legend. We used the VWF115 that does not have the SUMO and HSV tags to establish that the SUMO and HSV tags did not themselves have any effects upon proteolysis.

The Fig 1 legend now reads

“d) VWF A2 domain fragments that were used in this study. VWF96 is a 96 amino acid A2 domain fragment (Gly¹⁵⁷³-Arg¹⁶⁶⁸) with a 13kDa N-terminal SUMO tag and a short C-terminal HSV tag and that spans the ADAMTS₁₃ cleavage site and each of the Dis, Cys-rich and Spacer domain exosite binding regions. With the tags, VWF96 has a MW of ~32kDa. The regions deleted or mutated in each of the VWF fragments are highlighted. e) Schematic representation of the domain organization of ADAMTS₁₃. Domains are labelled. The C-terminal tail consisting of the TSP repeats 2-8 and CUB domains are depicted folding back to interact with the central Spacer domain. f) The proteolysis of VWF96 (32kDa) by ADAMTS₁₃ is compared to a previous VWF A2 domain fragment Glu¹⁵⁵⁴-Arg¹⁶⁶⁸, VWF115 (18kDa) that lacks SUMO and HSV tags, but contains a short N-terminal Xpress epitope and 6xHis tag. For this 0.5nM ADAMTS₁₃ was incubated with 3μM VWF substrate at 37°C. Sub-samples were taken at specified times and analyzed by SDS-PAGE to visualize proteolysis – note the smallest VWF115 cleavage product is not stained due to its small size. **These data reveal that the identity of the tags do not influence proteolysis.”**

*We have also added this information into the legend of **Supp Fig 1**, again to ensure clarity to the reader.*

“Equilibrium plate binding assay” a google search on this phrase indicates that it is only used in ADAMTS₁₃ literature and by a limited set of authors. The term should not be used; it suggest that the method employed measures equilibrium binding, which it does not, because it is sensitive to the off-rate.

The reviewer is, of course, correct. The assay is a binding assay performed in a plate – we have modified the text accordingly to avoid the confusion caused by the term equilibrium throughout the manuscript.

Several values mentioned in the text about “Equilibrium plate binding assay” appear to be inaccurate. For instance the increase in KD(app) for VWF96-Cys is 16-fold not 15-fold, whereas the increase for VWF-Dis binding is 15-fold not 14-fold.

*We apologise for these errors and thank the reviewer for highlighting our mistakes. We have modified the text on **page 6** accordingly to correct these errors.*

Page 7 half way and legend of Figure 6: “The electrostatic charged surface...” is incorrectly phrased. What is displayed in figure 6 is the electrostatic potential at the molecular surface.

*We agree that this is not written appropriately and have changed the sentences on **page 7** and in the **Fig 6 legend** such they are now correctly phrased.*

Page 8: In the enumeration of differences between ADAMTS₁₃ and ADAMTS_{1, 4 and 5}. Point (iv) the alpha-helix is deleted probably should read “the alpha-helix is not formed”, because the sequence alignment in Fig 7 suggests that the residues that form the helix are present in ADAMTS₁₃.

*Once again the reviewer is correct, and we have rewritten this on **page 8** to reflect that the α -helix is not formed rather than missing. In **Fig 7** the label of ‘missing α -helix is replaced by ‘ α -helix-less loop’.*